# `PubSub-VFL`: Towards Efficient Two-Party Split Learning in Heterogeneous Environments via Publisher/Subscriber Architecture

**Yi Liu[1], Yang Liu[2,*], Leqian Zheng[1], Jue Hong[2], Junjie Shi[2], Qingyou Yang[2], Ye Wu[2], and Cong Wang[1,*]**

[1]Department of Computer Science, City University of Hong Kong
[2]ByteDance Inc.
{yiliu247-c, leqizheng2-c}@my.cityu.edu.hk, congwang@cityu.edu.hk
{liuyang.fromthu, tanzhuo.107, shijunjie.george}@bytedance.com
{yangqingyou, wuye.2020}@bytedance.com

## Abstract

With the rapid advancement of the digital economy, data collaboration between organizations has become a well-established business model, driving the growth of various industries. However, privacy concerns make direct data sharing impractical. To address this, Two-Party Split Learning (a.k.a. Vertical Federated Learning (VFL)) has emerged as a promising solution for secure collaborative learning. Despite its advantages, this architecture still suffers from low computational resource utilization and training efficiency. Specifically, its synchronous dependency design increases training latency, while resource and data heterogeneity among participants further hinder efficient computation. To overcome these challenges, we propose `PubSub-VFL`, a novel VFL paradigm with a Publisher/Subscriber architecture optimized for two-party collaborative learning with high computational efficiency. `PubSub-VFL` leverages the decoupling capabilities of the Pub/Sub architecture and the data parallelism of the parameter server architecture to design a hierarchical asynchronous mechanism, reducing training latency and improving system efficiency. Additionally, to mitigate the training imbalance caused by resource and data heterogeneity, we formalize an optimization problem based on participants' system profiles, enabling the selection of optimal hyperparameters while preserving privacy. We conduct a theoretical analysis to demonstrate that `PubSub-VFL` achieves stable convergence and is compatible with security protocols such as differential privacy. Extensive case studies on five benchmark datasets further validate its effectiveness, showing that, compared to state-of-the-art baselines, `PubSub-VFL` not only accelerates training by $2 \sim 7\times$ without compromising accuracy, but also achieves a computational resource utilization rate of up to 91.07%.

## 1 Introduction

In the digital economy, data has become a crucial resource driving technological advancements in sectors like autonomous driving [1], healthcare [2], and e-commerce [3]. In this context, enterprises are increasingly collaborating to aggregate diverse data sources to train Machine Learning (ML) models, improving user experience and efficiency [4, 5]. However, centralized data collection poses significant privacy risks and potential breaches [6]. Additionally, regulations like the General Data Protection Regulation (GDPR) [7] impose strict limits on data aggregation and usage, requiring explicit consent and robust safeguards.

---

[*]Cong Wang and Yang Liu are the corresponding authors.

39th Conference on Neural Information Processing Systems (NeurIPS 2025).

To address the above privacy concerns, enterprises generally deploy Two-Party Split Learning [8–10] (a.k.a. Vertical Federated Learning (VFL) [11–14]) to collaboratively train ML models without accessing the original data, as shown in Fig. 1. Specifically, in VFL, the data is partitioned vertically, such that each party stores data corresponding to the same sample IDs but with different feature spaces [13, 15, 16]. This allows for a comprehensive analysis through a secure exchange of intermediate results, protecting sensitive data while facilitating joint model training [11, 17]. A notable example is the cooperation between banks and insurance companies aiming to train a model to predict a customer's credit score, a common interest for both entities. Each party holds data on the same individuals but in different feature spaces: banks maintain financial transaction records, while insurance companies retain car accident reports. Due to privacy concerns, regulatory constraints (such as GDPR and HIPAA [18]), and communication network limitations, these features must remain localized within each party. In such scenarios, the VFL approach becomes essential. Therefore, many efforts [19, 2, 20] have been invested in VFL research to further unlock the commercial value of data.

In SL-based VFL[2], each party trains a partial deep network up to a specific layer known as the cut layer (called *bottom model*), which maps raw data features into meaningful vector representations or *embeddings* for prediction tasks. These embeddings are securely transmitted (using methods like differential privacy [21], homomorphic encryption [4], or secure two-party computation [2]) to the *active party*, which holds the labeled data. The active party completes the training using the remaining part of the network (called *top model*) without accessing the raw data of the other parties, thus completing a forward propagation round. Subsequently, gradients are back-propagated from the final layer of the top model to the cut layer. Only the gradients at the cut layer are sent back to

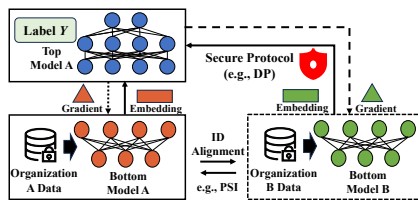

Figure 1: Overview of the split learning (SL)-based VFL framework.

the *passive party*, who then continues the backpropagation process locally. This iterative process is repeated until the model converges.

To reduce training costs and resource consumption, researchers and developers have been dedicated to designing efficient VFL architectures to enhance computational resource utilization and improve training efficiency. From a system-level perspective, previous works such as FATE [22] and PaddleFL [23] have focused on achieving efficient data parallelism by employing Parameter Server (PS) [24] architectures, thereby significantly improving computational resource utilization. At the algorithmic level, considerable effort has been devoted to developing effective asynchronous VFL training protocols to optimize computational efficiency [25]. However, these methods still face limitations. Some approaches may not fully address computational bottlenecks, while others might fail to maximize computational efficiency under resource and data heterogeneity conditions. We highlight the reasons behind the above design deficiencies as follows:

- **Neglect of the Decoupling of the System.** Since the VFL with PS setup is still synchronous and tightly coupled, there is a waiting bottleneck between workers on different parties [26], where they either wait for each other or have computational dependencies (see Fig. 6 in Appendix B). Introducing an asynchronous mechanism could potentially improve system decoupling. However, the unique characteristics of the VFL ID alignment [27] present significant challenges in implementing such a mechanism directly (see the detailed analysis in Appendix A).
- **Neglect of the Resource and Data Heterogeneity.** In VFL, computational resource and data feature dimensions often vary significantly among parties, leading to imbalances in computing time [28, 29]. Existing methods typically focus on incrementally improving the computational efficiency of individual parties, overlooking these discrepancies. Thus, the overall computational resources are underutilized due to the lack of a holistic approach to addressing this issue.

To address the limitations of existing approaches, we propose an efficient VFL framework named PubSub-VFL, designed to achieve better computational resource utilization and improve training efficiency. PubSub-VFL combines the loose coupling advantages of the Publisher/Subscriber (Pub/Sub) architecture with the data parallelism benefits of the PS architecture, enabling flexible asynchronous communication and significantly enhancing system throughput. To further boost computational efficiency and ensure convergence, we design an adaptive semi-asynchronous mechanism within the PS, forming a hierarchical asynchronous mechanism in conjunction with the Pub/Sub asynchrony outside

---

[2]For the sake of convenience, the term "VFL" is used in the context to refer to the term "SL-based VFL".

the PS. Additionally, to tackle load imbalance issues caused by resource and data heterogeneity, we develop an optimization model that leverages system profile information to determine optimal hyper-parameters, thereby further improving resource utilization. Extensive evaluations on five benchmark datasets demonstrate that PubSub-VFL achieves a $2 \sim 7\times$ improvement in computational efficiency over state-of-the-art baselines while maintaining model accuracy.

## 2 Related Work

**Vertical Federated Learning with PS.** The integration of PS [24] architectures into (V)FL systems has garnered attention due to its potential to improve scalability and flexibility. Early efforts such as [30] explored using PS architecture to achieve efficient aggregation and distribution of model parameters, simplifying the coordination of multiple participants, but these were primarily focused on horizontal FL settings. More recently, some solutions demonstrated the applicability of PS architectures in VFL contexts, showing improvements in resource utilization and dynamic participant management [26, 22, 23]. Castiglia *et al.* in [31] extended this approach by introducing hierarchical PS architectures, allowing for better load balancing and reduced latency in VFL. However, these solutions still grapple with issues such as decoupling from the unique step in VFL (i.e., ID alignment) to further unlock the potential of PS.

**Asynchronous Vertical Federated Learning.** Asynchronous training [32–34] methods have gained traction in VFL to address the inefficiencies caused by straggler effects and the need for strict synchronization. Asynchronous VFL (AVFL) enables participants to train and exchange intermediate results independently, significantly improving system throughput and scalability. Recent works, such as those by [34], have demonstrated the efficacy of AVFL in mitigating delays caused by slow participants, thereby enhancing the overall efficiency of the learning process. Nevertheless, asynchronous methods introduce new complexities, including staleness in gradients and potential divergence of models. To tackle these issues, recent research has focused on developing consistency models [35, 36] and adaptive asynchronous strategies [37]. We summarize the comparison between existing methods and PubSub-VFL in Table 5 in Appendix A.

## 3 Problem Formulation

We consider a realistic scenario where two organizations collaboratively train an ML model to perform a prediction task. We then formalize a framework for VFL, with the objective of maximizing the utilization of computing resources between two parties. In VFL, data is vertically partitioned such that each party holds different features for the same set of samples. Specifically, Active Party $P_a$ possesses dataset $D_1 = \{(\mathbf{x}_i^a, y_i)\}_{i=1}^n$, where $\mathbf{x}_i^a \in \mathbb{R}^{d_a}$ represents its feature vectors and $y_i \in \mathbb{R}$ denotes the labels, while Passive Party $P_p$ holds dataset $D_2 = \{\mathbf{x}_i^p\}_{i=1}^n$, with $\mathbf{x}_i^p \in \mathbb{R}^{d_p}$. Note that the features of the two parties are disjoint. The model architecture is split into two parts: the top model and the bottom model, where the bottom models $f_a(\cdot)$ and $f_p(\cdot)$ are held by $P_a$ and $P_p$, respectively, while the top model $g(\cdot, \cdot)$ is held by $P_a$ which holds the label. Before training begins, $P_a$ and $P_p$ must identify data samples with matching identifiers (such as ID). To preserve the privacy of both parties, Private Set Intersection (PSI) [38] technique is typically employed to securely obtain the "shared" dataset without revealing any private information. During the forward pass, $P_p$ computes an intermediate representation $z_i^p = f_p(\mathbf{x}_i^p)$ (i.e., embeddings) and securely sends it to $P_a$ by using methods like DP protocol, which then computes $z_i^a = f_a(\mathbf{x}_i^a)$ and aggregates these representations using $g(\cdot)$ to produce $\hat{y}_i$, i.e., $\hat{y}_i = g(f_a(\mathbf{x}_i^a), f_p(\mathbf{x}_i^p))$. The loss function $\mathcal{L}(\hat{y}_i, y_i)$ is calculated at $P_a$. For example, for a classification task, the loss function can be the Cross-Entropy Loss:

$$\mathcal{L}(\hat{y}_i, y_i) = -\frac{1}{n} \sum_{i=1}^n \left( y_i \log(\hat{y}_i) + (1 - y_i) \log(1 - \hat{y}_i) \right). \tag{1}$$

In the backward pass, $P_a$ computes gradients $\nabla_{z_i^a} \mathcal{L}$ and $\nabla_{\theta_1} \mathcal{L}$, sending $\nabla_{z_i^p} \mathcal{L}$ back to $P_p$, which uses it to compute $\nabla_{\theta_2} \mathcal{L}$. Both parties update their local model parameters using gradient descent:

$$\theta_1 \leftarrow \theta_1 - \eta \nabla_{\theta_1} \mathcal{L}, \theta_2 \leftarrow \theta_2 - \eta \nabla_{\theta_2} \mathcal{L}, \tag{2}$$

where $\eta$ is the learning rate, and $\theta_1$ and $\theta_2$ are the network parameters of $f_1$ and $f_2$, respectively.

To maximize the utilization of computational resources, we need to balance the computational load between the two parties. Let $Cost_A$ denote the computational cost of $P_a$ for processing $f_a$ and

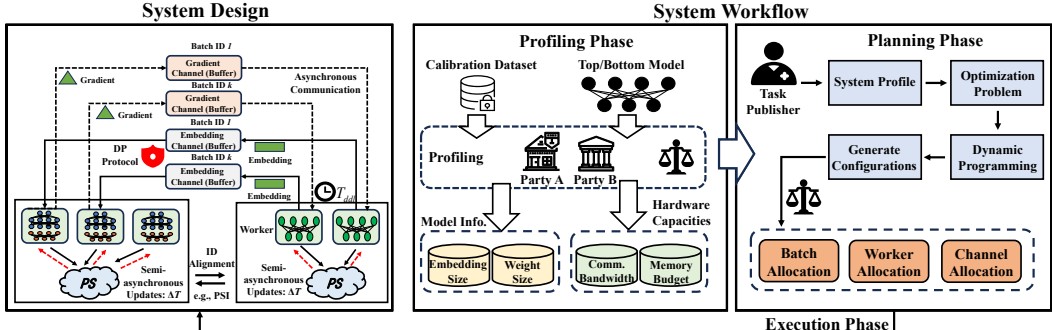

Figure 2: System architecture and workflow of `PubSub-VFL`.

$Cost_P$ denote the computational cost of $P_p$ for processing $f_p$. We use $T_A$ to denote the total time taken by $P_a$ for one iteration and use $T_P$ to denote the total time taken by $P_p$ for one iteration. The goal is to minimize the maximum time taken by either party, i.e., minimize $\max(T_A, T_P)$. Assuming that the communication overhead is not negligible, we can write:

$$T_A = Cost_A + t_g, T_P = Cost_P + t_e, \tag{3}$$

where $t_g$ is time for sending gradient $\nabla_{\theta_2}\mathcal{L}$ and $t_e$ is time for sending $z_i^p$. To balance the load, we aim to equalize $T_A$ and $T_P$: $T_A \approx T_P$. This can be achieved by adjusting the training efficiency of $f_a$ and $f_p$ or dynamically allocating computing resources. For example, if $Cost_A > Cost_P$, we can improve the training efficiency of $f_a$ or increase the training efficiency of $f_p$ to balance the workload. The formal objective is to minimize the maximum time taken by either party while ensuring convergence of the model:

$$\min \max(T_A, T_P), \tag{4}$$

subject to: $\frac{1}{n}\sum_{i=1}^{n}\mathcal{L}(\hat{y}_i, y_i) \leq \kappa$, where $\kappa$ is a small tolerance for the loss.

**Discussion.** Optimizing Eq. (4) is a challenging task due to two key factors: 1) **Resource and Data Heterogeneity.** In practice, $P_a$ and $P_p$ often differ significantly in their computing resources and data dimensions. Additionally, $P_p$'s computational overhead in VFL is notably lower than $P_a$'s, as $P_p$ does not need to perform a backward pass of the top model. Therefore, it is essential to balance resource allocation while accounting for the computational complexities of both parties. 2) **Privacy Restrictions.** Due to privacy concerns, it is not feasible to manage and control resource allocation in a centralized manner. This limitation distinguishes the problem from traditional resource scheduling in VFL, requiring more nuanced approaches to ensure privacy while optimizing resource utilization.

## 4 System and Algorithm Design

### 4.1 System Design

**Key Idea.** To address the scarecrow solution's limitations (more details can be found in Appendix A), our key idea is to *decouple the data ID alignment task from the training tasks of workers across different parties.* This approach allows workers to focus solely on local training without the need to ensure data ID alignment beforehand. To implement this, we introduce the *Publisher/Subscriber* (Pub/Sub) [39] architecture, which effectively separates the training process from the data ID matching task, enabling asynchronous operations while maintaining the necessary alignment. By doing so, we can eliminate waiting delays, improve system concurrency, and ensure seamless data ID alignment without burdening the workers with additional coordination tasks. More information about the Pub/Sub can be found in the Appendix B.

To this end, we design the `PubSub-VFL` system to support efficient and scalable VFL. As illustrated in Fig. 2, we introduce two types of communication channels: *embedding channels* and *gradient channels*, responsible for transmitting embeddings and gradients, respectively. To decouple data alignment from model training, we assign a unique batch ID to each training batch. This batch ID is used to label both embedding and gradient channels, enabling precise coordination of intermediate

results across parties. Each worker sends its outputs to the appropriate channel based on the batch ID, avoiding synchronization delays. Given a total of $n$ training samples and a batch size $B$, the system maintains $\lceil n/B \rceil$ embedding and gradient channels. This design ensures consistent data alignment while supporting asynchronous training, thereby improving efficiency. In a more practical scenario, computationally efficient workers may generate excessive embeddings or gradients, leading to channel congestion, which can impede model convergence or even cause training failures. To address this, we propose two mechanisms:

- **Buffer Mechanism:** Each channel buffer can store up to $p$ embeddings and $q$ gradients, with each entry timestamped. When the buffer reaches its capacity, it discards the oldest embedding or gradient based on a First-In-First-Out (FIFO) [40] principle to prevent stale updates from affecting the training process.
- **Waiting Deadline Mechanism:** If a subscribing worker does not receive the embedding or gradient from the publishing worker within a predetermined time $T_{ddl}$, it discards the current batch and notifies the other party to proceed with the next batch. The system then reassigns the batch to any available pair of workers for retraining, ensuring the continuity of the training process and mitigating delays caused by congestion.

**Intra-party Semi-asynchronous Mechanism.** Building on the Pub/Sub architecture, we achieve inter-party asynchronous communication, effectively eliminating worker-side waiting delays. To further improve computational efficiency within each party (i.e., between the PS and its workers), we extend this design with an intra-party asynchronous mechanism. However, this hierarchical asynchrony can hinder model convergence if not properly controlled. To mitigate this, we propose a dynamic semi-asynchronous mechanism that adaptively regulates the synchronization interval based on training feedback. Specifically, the synchronization interval $\Delta T_t$ decreases as the model approaches the target accuracy, striking a balance between computation speed and convergence stability. The interval $\Delta T_t$ is defined as:

$$\Delta T_t = \left\lceil \frac{\Delta T_0}{2} \cdot \tanh\left(\frac{2 \cdot t}{\Delta T_0} - 2\right) + \frac{\Delta T_0}{2} \right\rceil, \tag{5}$$

where $\Delta T_0$ is the initial asynchronous interval, $t$ is the current training epoch, and $\tanh(\cdot)$ is the hyperbolic tangent function. Initially, when the model is far from the target accuracy, $\Delta T_t$ is small, allowing the model to achieve stable learning. As the accuracy increases, the interval increases and the synchronization frequency is reduced to fine-tune the model and ensure faster convergence.

**Differential Privacy Protocol.** To prevent the embedded information sent by $P_p$ from being inferred by attackers (such as labels or features), specific perturbations must be added to the embeddings. To balance privacy and utility effectively, we adopt Gaussian Differential Privacy (GDP) (see Appdenix C for more details) as outlined in [21, 33] to safeguard the embedding information. In addition, we prove in Appendix D that PubSub-VFL integrated GDP protocol can still converge stably.

## 4.2 System Profiling

**Key Idea.** PubSub-VFL effectively integrates Pub/Sub and PS architectures to harness the computational capabilities of participating parties, substantially improving training efficiency. Nevertheless, the system remains susceptible to latency caused by resource and data heterogeneity. A key challenge lies in optimizing resource allocation collaboratively without violating privacy constraints. To address this, we propose a privacy-preserving parameter optimization strategy that determines optimal configurations, e.g., the number of workers, batch size, and core allocation, based on each party's system profile, including model characteristics and hardware capabilities. By adapting these parameters to individual constraints, the system achieves balanced workload distribution, reduced latency, and improved overall efficiency, all while preserving data privacy.

To achieve this goal, we first model the computation and communication delays of both the active and passive parties in the designed PubSub-VFL system. For the $P_a$, we denote the number of workers as $w_a \in [P, Q]$, batch size as $B$, and total computing cores as $C_a$. For the $P_p$, the number of workers is $w_p \in [M, N]$, batch size as $B$, and total computing cores as $C_p$. Let $\{c_{a,i}\}_{i=1}^{w_a}$ (where $\sum_{i=1}^{w_a} c_{a,i} \leq C_a$, be the CPU cores allocated to each of the $w_a$ and $\{c_{p,j}\}_{j=1}^{w_p}$ (where $\sum_{j=1}^{w_p} c_{p,j} \leq C_p$) be the CPU cores allocated to each of the $w_p$. Therefore, the computation delay

of the forward pass of both parties can be formally defined as follows:

$$T_f^{(a)}(B) = \frac{\lambda_a B^{\gamma_a}}{\sum\limits_{j=1}^{w_a} c_{a,j}}, T_f^{(p)}(B) = \frac{\lambda_p B^{\gamma_p}}{\sum\limits_{i=1}^{w_p} c_{p,i}},$$ (6)

where $T_f^{(a)}(B)$ is the time for forward pass of the $P_a$ on a batch of size $B$, $T_f^{(p)}(B)$ is the time for forward pass of the $P_p$ on a batch of size $B$, $\gamma_a$, $\lambda_a$, $\gamma_p$, and $\lambda_p$ are the proportionality constants. For simplicity, if all $w_p$ workers are assigned equally, it might become $T_f^{(a)}(B) = \frac{\lambda_a B^{\gamma_a} w_a}{C_a}$ and $T_f^{(p)}(B) = \frac{\lambda_p B^{\gamma_p} w_p}{C_p}$. Similarly, let $\beta_a$ and $\beta_p$ be the constant for the backward pass. Thus, we have:

$$T_b^{(a)}(B) = \frac{\varphi_a B^{\beta_a} w_a}{C_a}, T_b^{(p)}(B) = \frac{\varphi_p B^{\beta_p} w_p}{C_p}.$$ (7)

We give the forward and backward propagation time of the top model part of $P_a$ as follows:

$$T_{top}^{(a)}(B) = \frac{\lambda_a' B^{\gamma_a'} w_a}{C_a} + \frac{\varphi_a' B^{\beta_a'} w_a}{C_a}.$$ (8)

Next, we consider the communication delay within the system pipeline. Let $E$ denote the size of the embedding sent by the $P_p$ and $G$ the size of the gradient sent by the $P_a$. Each iteration involves two primary communications: $P_p$ sends the embedding to $P_a$, and $P_a$ sends the gradient back to $P_p$. Therefore, the total communication delay for each iteration can be expressed as:

$$T_{emb} = \frac{E}{B_b}, T_{grad} = \frac{G}{B_b}.$$ (9)

where $B_b$ denotes the bandwidth capacities of the system. Since the semi-asynchronous aggregation within the PS is performed internally within each participant, the communication delays from this step can be ignored, as they are generally fixed constants.

**Remark on Pipelining/Asynchronous Pub/Sub.** With a Pub/Sub architecture, one can sometimes overlap the next batch's forward pass at the $P_p$ with the $P_a$'s backward pass from the previous batch. In a fully pipelined system with enough buffering, the overall iteration time can be lower than the naive sum above. Nonetheless, many system analyses still approximate iteration time by a "critical path" sum or a maximum of partial sums [41, 42]. For simplicity, we continue with the additive formula here, but in practice, pipelining would reduce that total somewhat.

### 4.3 System Planning Phase

**Optimization Problem Formulation.** Due to privacy constraints and restricted network access, participants cannot collaboratively execute fine-grained pipeline operations. To overcome this limitation, we estimate the computation and communication times in the Pub/Sub setting using observations from a synchronous baseline, as illustrated in Fig. 2. These estimations enable the construction of an optimization model to determine the optimal initialization hyperparameters, specifically, the number of workers and batch size. This model aims to balance computation and communication costs while maintaining system efficiency under privacy and network constraints. Based on Eq. (4), the formal expressions for $T_A$ and $T_P$ can be rewritten as:

$$T_A = T_f^{(a)} + T_b^{(a)} + T_{top}^{(a)} + T_{grad}, T_P = T_f^{(p)} + T_b^{(p)} + T_{emb}.$$ (10)

Then, we rewrite Eq. (4) as follows:

$$\min \max(T_f^{(a)} + T_b^{(a)} + T_{top}^{(a)} + T_{grad}, T_f^{(p)} + T_b^{(p)} + T_{emb}).$$ (11)

In Eq. (11), we want to choose the optimal hyperparameters, i.e., $w_a \in \{P, P+1, \ldots, Q\}$, $w_p \in \{M, M+1, \ldots, N\}$, an integral (or real) batch size $1 \le B \le B_{max}$ (bounded above by some feasible maximum, i.e., memory constraints), to optimize the goal. We assume that each party has its own memory constraint per worker. Thus, we follow [41] to specify the memory usage functions as

$$M_A(B) = M_{A0} + \rho_A B^\chi, M_P(B) = M_{P0} + \rho_P B^\chi,$$ (12)

where $M_{A0}$ and $M_{P0}$ are the base memory consumptions at the active and passive parties respectively, and $\rho_A, \rho_P$ (with exponent $\chi$) capture the extra memory needed per worker as a function of the minibatch size $B$. If the maximum available memory per worker at the $P_a$ is $\bar{M}_A$ and at the $P_p$ is $\bar{M}_P$, then the memory constraints become $B \leq \left( \frac{\bar{M}_A - M_{A0}}{\gamma_A} \right)^{1/\chi}, B \leq \left( \frac{\bar{M}_P - M_{P0}}{\gamma_P} \right)^{1/\chi}$. Thus, we define the overall feasible maximum batch size due to memory as

$$B_{\max} = \min \left\{ \left( \frac{\bar{M}_A - M_{A0}}{\rho_A} \right)^{1/\chi}, \left( \frac{\bar{M}_P - M_{P0}}{\rho_P} \right)^{1/\chi} \right\}. \tag{13}$$

We assume that the candidate minibatch sizes are taken from a discrete set, i.e., $\mathcal{B} = \{B_1, B_2, \ldots, B_R\}$, but only those values that satisfy $B \leq B_{\max}$ are feasible. Since both parties must finish their computations before proceeding to the next iteration, the overall per-iteration delay is the maximum of the two computation delays plus the communication delay:

$$\min \mathcal{O}(w_A, w_P, B) = \min_{w_a, w_p, B \leq B_{\max}} \left\{ \max \left( T_f^{(a)} + T_b^{(a)} + T_{top}^{(a)}, T_f^{(p)} + T_b^{(p)} \right) + \frac{E + G}{B_b} \right\}. \tag{14}$$

**Dynamic Programming Algorithm Design.** Because the decision space (over $w_a$, $w_p$, and $B$) is discrete, we now describe a dynamic programming approach to search for the optimal configuration. We define a dynamic programming state by the triplet $(i, j, r)$, where, $i$ indexes the candidate active worker count: $w_a = M + i - 1$ for $i = 1, 2, \ldots, (N - M + 1)$, $j$ indexes the candidate passive worker count: $w_p = P + j - 1$ for $j = 1, 2, \ldots, (Q - P + 1)$, and $r$ indexes the candidate minibatch size: $B = B_r$ for $r = 1, 2, \ldots, R$, with the additional constraint $B_r \leq B_{\max}$. The cost associated with state $(i, j, r)$ is defined as

$$\text{Cost}(i, j, r) = \max \left\{ \frac{(\lambda_a B_r^{\gamma_a} + \lambda_a' B_r^{\gamma_a'} + \varphi_a B_r^{\beta_a} + \varphi_a' B_r^{\beta_a'})(M + i - 1)}{C_a}, \right.$$
$$\left. \frac{(\lambda_p B_r^{\gamma_p} + \varphi_a B_r^{\beta_p})(P + j - 1)}{C_p} \right\}. \tag{15}$$

The objective is to find the state $(i^*, j^*, r^*)$ that minimizes this cost. The above dynamic programming solution (Algo. 2) and the pseudo code of PubSub-VFL (Algo. 1) can be found in the Appendix E.

## 5 Experiment

### 5.1 Experiment Setup

To evaluate the performance of our PubSub-VFL system, we conduct extensive experiments on five datasets. All experiments are developed using Python 3.9 and PyTorch 1.12 and evaluated on a server with an INTEL(R) XEON(R) GOLD 6530 (64-core CPU).

**Datasets.** We evaluate PubSub-VFL on four public benchmark datasets (see Table 6 in Appendix F) spanning both regression and classification tasks, along with a large-scale synthetic dataset. For regression, we use the Energy [43] (19,735 samples, 27 features) and Blog [44] (60,021 samples, 280 features) datasets. For

Table 1: Accuracy comparison results.

| Dataset | Metric | VFL | VFL-PS | AVFL | AVFL-PS | Ours |
|---|---|---|---|---|---|---|
| Energy | RMSE | 84.58 | 84.44 | 85.41 | 85.39 | 85.64 |
| Blog | RMSE | 23.20 | 23.12 | 23.38 | 23.45 | **22.34** |
| Bank | AUC | 94.54 | 94.13 | 94.12 | 94.16 | **96.54** |
| Credit | AUC | 81.90 | 81.34 | 80.83 | 80.34 | **82.34** |
| Synthetic | AUC | 91.27 | 91.31 | 90.97 | 91.21 | **92.87** |

classification, we adopt the Bank [45] (40,787 samples, 48 features) and Credit [46] (30,000 samples, 23 features) datasets. To assess scalability, we generate a synthetic dataset with 1 million samples and 500 features using Scikit-learn [47]. Each dataset is split into 70% training and 30% testing, with training data approximately evenly distributed between two parties. To simulate feature heterogeneity, we vary the number of features assigned to each party.

**Models.** For the top model, we use a Multi-Layer Perceptron (MLP) with two layers. For the bottom model, we use two models of different sizes, namely a ten-layer MLP and a ResNet [48], which can verify the performance of PubSubVFL under different model sizes.

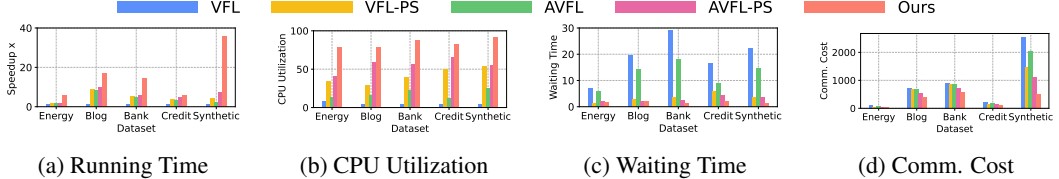

(a) Running Time  (b) CPU Utilization  (c) Waiting Time  (d) Comm. Cost

Figure 3: Comparison with existing baselines in computation and communication efficiency.

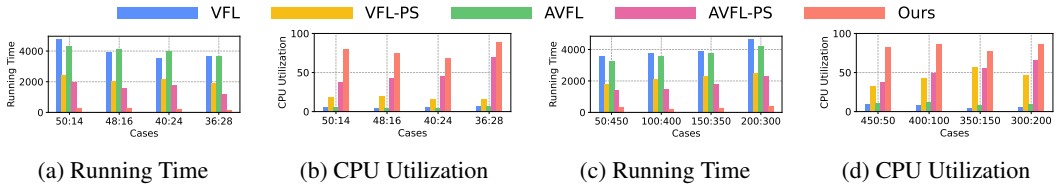

(a) Running Time  (b) CPU Utilization  (c) Running Time  (d) CPU Utilization

Figure 4: Comparison with existing baselines on computation efficiency in resource and data heterogeneous scenarios.

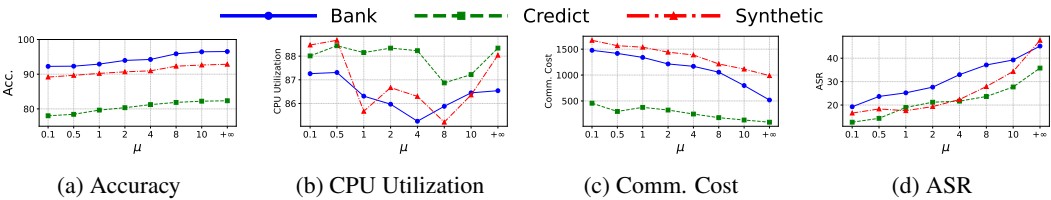

(a) Accuracy  (b) CPU Utilization  (c) Comm. Cost  (d) ASR

Figure 5: The impact of privacy budget on the performance, efficiency, and security of `PubSub-VFL`.

Table 2: Effect of the number of workers.

| # of Workers | 4 | 5 | 8* | 10 | 20 | 30 | 50 |
|---|---|---|---|---|---|---|---|
| Acc.(%) | 92.13 | 92.05 | **92.06** | 92.28 | 92.00 | 92.36 | 92.21 |
| Time (s) | 712.78 | 805.90 | **668.11** | 885.01 | 1420.32 | 1067.57 | 1661.74 |
| CPU (%) | 67.52 | 63.30 | **88.04** | 76.18 | 42.77 | 40.78 | 45.12 |
| Waiting (s) | 1.4686 | 1.9273 | **1.5288** | 3.461 | 8.088 | 9.687 | 19.843 |
| Comm. (MB) | 878.91 | 1098.63 | **888.77** | 1318.36 | 1867.68 | 1538.09 | 2197.27 |

Table 3: Effect of the different batch size.

| Batch Size | 16 | 32 | 64 | 128 | 256 | 512 | 1024 |
|---|---|---|---|---|---|---|---|
| Acc.(%) | 91.70 | 92.06 | 91.75 | 92.63 | **92.67** | 92.36 | 92.21 |
| Time (s) | 987.64 | 668.11 | 344.76 | 124.01 | **92.54** | 578.69 | 865.74 |
| CPU (%) | 48.64 | 88.04 | 90.12 | 89.97 | **91.07** | 84.47 | 52.67 |
| Waiting (s) | 1.087 | 1.5288 | 1.688 | 1.263 | **1.1389** | 1.324 | 1.789 |
| Comm. (MB) | 1298.32 | 888.77 | 329.59 | 439.45 | **439.45** | 736.89 | 1070.36 |

**Parameters.** For a series of constants, we set $\Delta T_0 = 5$, $T_{ddl} = 10s$, $p = 5$, $q = 5$. For the constants in the optimization model, we determined them through empirical experiments (see the Appendix H for details). In addition, we set the learning rate to 0.001, the number of workers to $w_a/w_p \in [2, 50]$, the batch size $B \in \{16, 32, 64, 128, 256, 512, 1024\}$, and $C_a + C_p = 64$.

**Baselines.** In this paper, we adopt the following baselines: 1) *Pure VFL:* This is a classic VFL architecture that does not involve the PS architecture or asynchronous mechanisms. 2)*VFL with PS:* This is the most widely adopted VFL architecture in the industry, implemented in mature frameworks such as FATE [22] and PaddleFL [23]. By leveraging the PS architecture, it enhances computational resource utilization and efficiency, enabling more effective parallel processing. 3) *Asynchronous VFL:* Building on traditional VFL, developers can integrate asynchronous mechanisms [32, 34] to implement Asynchronous VFL (AVFL), enhancing system efficiency by reducing idle time and improving parallelism. 4) *Asynchronous VFL with PS:* Building on AVFL, developers can integrate the PS mechanism [26] to further enhance VFL efficiency. Notably, the asynchronous implementation in this architecture is achieved through inter-party communication between parties.

**Evaluation Metrics.** For the four public benchmark datasets, we evaluate classification tasks using the Area Under the ROC Curve (AUC) and regression tasks using the Root Mean Square Error (RMSE). For the synthetic dataset, we use AUC as the evaluation metric for classification tasks. To fairly assess computational resource utilization, we measure running time, CPU utilization, and waiting time/epoch. Additionally, we record communication cost to compare the communication efficiency of different methods. Results of additional experiments can be found in Appendix H.

## 5.2 Numerical Results

**System Performance.** We evaluate the performance of `PubSub-VFL` and baseline methods on five datasets across classification and regression tasks. To ensure a balanced allocation of resources and data, we evenly distribute CPU cores and feature sizes between the two parties. Additionally, to assess the impact of different bottom model architectures, we employ both MLP and ResNet models. We report the best performance of each method under optimal hyperparameter configurations in Table 1 (small size model) and Table 7 (large size model). The experimental results demonstrate that `PubSub-VFL` achieves accuracy comparable to or even surpassing baseline methods on the Bank, Credit, and Synthetic datasets. This confirms that integrating the Pub/Sub architecture and semi-asynchronous mechanism does not compromise system performance or convergence.

**Comp. & Comm. Costs.** We evaluate the computational and communication efficiency of `PubSub-VFL` against baseline methods. Using our strategy, we set the hyperparameters to $B = 256$, $w_a = 8$, and $w_p = 10$. For computational efficiency, we measure total running time, CPU utilization, and per-epoch waiting time required to reach a target accuracy of 91%. Communication efficiency is assessed via the total communication cost. Experiments on the synthetic dataset show that `PubSub-VFL` significantly outperforms all baselines in both computation and resource utilization. As shown in Fig. 3, `PubSub-VFL` achieves a $7\times$ reduction in running time and 35% higher CPU utilization compared to the best-performing baseline, AVFL-PS. These gains stem from reduced worker idle time and improved parallelism. Moreover, the hierarchical asynchronous mechanism enhances convergence efficiency, leading to lower communication cost than other methods.

**Resource and Data Heterogeneity Scenarios.** We evaluate the computational efficiency of `PubSub-VFL` and the baseline methods under varying resource allocation and feature size distribution scenarios. For the resource heterogeneous scenario, we set different CPU core ratios between $P_a$ and $P_p$: 50:14, 48:16, 40:24, and 36:28 (where the first value represents $P_a$'s CPU cores). For the data heterogeneous scenario, we set different feature size ratios: 50:450, 100:400, 150:350, and 200:300. In each scenario, we apply our optimization method to determine the best hyperparameters and configure them for `PubSub-VFL`. The experimental results, shown in Fig. 4, reveal that in the resource heterogeneous scenario, imbalanced computational efficiency between parties significantly increases waiting time and decreases CPU utilization in baseline methods. Specifically, the CPU utilization of `PubSub-VFL` is still as high as 87.42% when the CPU core ratio is 50:14, while that of AVFL-PS is only 42.12%. This happens because resource disparities exacerbate computational imbalances, further extending training latency. In contrast, `PubSub-VFL` effectively balances computational efficiency, reducing running time and maintaining high CPU utilization. In the data heterogeneity scenario, we observe similar trends. Additionally, we find that reducing the data dimension processed by $P_a$ can further decrease running time (as shown in Fig. 4 (c)–(d)). This is because it helps balance the computational load between both parties, aligning with our optimization model design approach.

**Security Performance Evaluation.** We evaluate it from two dimensions: system performance and defense against embedded inversion attacks [49]. We follow the above hyperparameter configuration to configure `PubSub-VFL`. For the privacy parameter, we set $\mu \in \{0.1, 0.5, 1, 2, 4, 8, 10, +\infty\}$.

*System Performance.* We evaluate the impact of $\mu$ on `PubSub-VFL` by recording its accuracy, CPU utilization, and communication cost on the Bank, Credit, and Synthetic datasets. The results, presented in Fig. 5, demonstrate that introducing the DP protocol has minimal effect on accuracy and CPU utilization, indicating that `PubSub-VFL` maintains its computational efficiency even with privacy protection. However, we observe a notable increase in communication cost due to the added DP noise, which leads to a slower convergence. Nevertheless, the results confirm that `PubSub-VFL` seamlessly integrates with security protocols while maintaining strong performance.

*Defend Against Embedding Inversion Attacks (EIA).* Similarly, we record the performance results (i.e., Attack Success Rate (ASR)) of `PubSub-VFL` against EIA (we adopt it in [49]) on these datasets (more details can be found in the Appendix G). The results are shown in Fig. 5, showing that the introduction of the DP protocol can help `PubSub-VFL` defend against EIA well.

**Parameter Sensitivity Evaluation.** We evaluate the impact of different numbers of workers and batch sizes on the performance of `PubSub-VFL` under the same setting. Specifically, we set $w_a = w_p \in \{4, 5, 8, 10, 20, 30, 50\}$ and $B \in \{16, 32, 64, 128, 256, 512, 1024\}$.

Table 4: Comparison of Different Methods on Various Datasets

| Method | Energy | Blog | Bank | Credit | Synthetic |
|---|---|---|---|---|---|
| All (PubSub-VFL) | 83.94 | 22.14 | 96.97 | 86.07 | 94.17 |
| w/o $T_{all}$ | 84.35 | 23.17 | 95.26 | 85.74 | 92.86 |
| w/o Dynamic Programming | 84.07 | 22.16 | 96.33 | 85.79 | 93.82 |
| w/o $\Delta T$ | 85.68 | 24.11 | 95.01 | 84.45 | 92.07 |
| w/o PubSub | 83.98 | 22.66 | 95.17 | 85.93 | 93.52 |
| w/o $T_{all}$ and $\Delta$ | 85.81 | 24.24 | 94.32 | 82.69 | 91.73 |
| VFL | 84.24 | 23.18 | 94.97 | 83.42 | 92.74 |
| VFL-PS | 86.14 | 23.07 | 94.74 | 85.44 | 92.67 |
| AVFL | 83.91 | 22.97 | 95.02 | 84.23 | 91.54 |
| AVFL-PS | 84.29 | 23.15 | 95.06 | 82.27 | 92.21 |

*Effect of the Numer of Workers.* We conduct experiments on the Synthetic dataset with $B = 32$, and the experimental results are shown in Table 2. The results show that simply increasing the parallel factor (i.e., $w$) does not always improve computational efficiency. For example, we find that when $w = 8$, its computation and communication efficiency is the highest. This is because a large parallel factor will lead to slower convergence.

*Effect of the Batch Size.* Similarly, we conduct experimens on the Synthetic dataset with $w_a = w_p = 8$. Table 3 records the impact of different $B$ on the performance of PubSub-VFL. We find that blindly increasing $B$ cannot always improve the computational efficiency of PubSub-VFL. For example, we find that when $B = 256$, its computation and communication efficiency is the highest. This is because too large a batch size will also lead to slower convergence. The above results verify that we need a suitable method to find the optimal hyperparameters in the VFL task.

**Ablation Studies.** We conduct comprehensive ablation studies following the above experimental setup, including the same model architectures, number of clients, and heterogeneous resource settings. Experiments are performed on the Energy, Blog, Bank, Credit, and Synthetic datasets to evaluate the individual contributions of PubSub-VFL's key components. Specifically, to assess the impact of the waiting deadline mechanism, we set the deadline to $T_{all} = 0s$, effectively disabling the mechanism. To evaluate the dynamic programming algorithm, we adopt a fixed worker allocation (i.e., equal numbers of workers on both party sides), removing the adaptive scheduling it enables. For the intra-party semi-asynchronous mechanism, we remove this component while retaining the PS architecture to isolate its effect. Finally, to study the role of the PubSub architecture, we replace it with the AVFL-PS architecture while keeping all other components unchanged. These ablation experiments allow us to disentangle the influence of each design choice on overall system performance. The experimental results are shown in the Table 4. The Waiting Deadline and Intra-party Semi-asynchronous Mechanisms are pivotal to PubSub-VFL's performance, effectively balancing synchronization and asynchrony to mitigate gradient staleness and ensure timely updates. Their removal incurs significant degradation, e.g., up to 2.10% AUC drop on Synthetic and 1.71% on Bank—highlighting their role in stable convergence. The PubSub architecture and dynamic programming contribute more modestly to performance but enhance robustness by alleviating resource heterogeneity and coordination overhead, improving stability under coupled training dynamics.

## 6  Conclusion

The paper addressed the problem of underutilization of computational resources in VFL by proposing a new framework named PubSub-VFL. Specifically, PubSub-VFL enhanced computational efficiency by leveraging a Pub/Sub architecture with hierarchical asynchronous mechanism. Furthermore, we theoretically prove the convergence of PubSub-VFL. It reduces training latency, improves resource utilization, and maintains strong convergence and privacy guarantees, achieving up to $2 \sim 7\times$ faster training and $\approx 35\%$ better resource efficiency than state-of-the-art baselines.

**Limitations.** One limitation of PubSub-VFL is that it only supports two-party learning and has not yet been able to support multi-party learning. In future exploration work, we will seek ways to support efficient multi-party learning.

## Acknowledgements

We thank all anonymous reviewers for their constructive comments. This work was supported in part by the Hong Kong Research Grants Council under Grants CityU 11218322, 11219524, R6021-20F, R1012-21, RFS21221S04, C2004-21G, C1029-22G, C6015-23G, and N_CityU139/21 and in part by the Innovation and Technology Commission (ITC) under the Joint Mainland-Hong Kong Funding Scheme (MHKJFS) under Grant MHP/135/23. This work was also supported by the InnoHK initiative, The Government of the HKSAR, and the Laboratory for AI-Powered Financial Technologies (AIFT).

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

# A  Scarecrow Solution

To optimize Eq. (4), a common approach is to implement a PS architecture within each party to achieve parallel processing. This architecture is widely adopted by industrial-level VFL frameworks like FATE [22] and PaddleFL [23]. The PS architecture in VFL typically involves each party setting up a PS and a set of workers, each running a VFL executor responsible for model training and security protocols. For a given task, the coordinator, responsible for scheduling system components, determines the parallel factor $\nu$, which specifies the number of workers to be created. It then instructs the agents of each party to initiate the PS and workers. This setup involves both inter-party Peer-to-Peer communication among workers from different parties, as well as intra-party communication between the PS and its workers, ensuring efficient coordination and execution of the VFL task [26].

Specifically, in each iteration, the $P_a$'s PS selects a batch and divides the instance IDs into $q$ subsets, such as ID $= \{\text{ID}_1, \dots, \text{ID}_q\}$. PS then broadcasts these subsets to the $P_p$'s PS, which distributes each subset $\text{ID}_j$ to its workers. This ensures that during execution, the PS and workers of each party are aligned, with each group of workers processing the same instances. The workers of $P_a$ and $P_p$ execute forward propagation concurrently, with the workers of $P_a$ completing the remaining forward propagation and performing backward propagation using the top model. Meanwhile, once the workers of $P_p$ receive the corresponding gradients, they carry out backward propagation to update their local models. Afterward, each worker uploads the updated model parameters to its respective PS. Finally, the PS aggregates these updates and broadcasts the refined model parameters to its workers, completing the iteration.

**Limitations.**  While deploying the PS architecture in VFL can significantly improve training efficiency (i.e., decrease $Cost_1$ and $Cost_2$), it still faces certain limitations. Firstly, due to disparities in computational overhead among the parties, the computation times in VFL are unbalanced. Since PS must ensure strict ID alignment, the faster worker in a pair of workers must wait for the slower one, or the training process will fail due to mismatched embeddings. Moreover, deploying a PS does not fully address the issues of resource and data heterogeneity between parties, as the PS architecture cannot facilitate resource collaboration between different parties. This limits its ability to optimize resource allocation and manage data heterogeneity effectively in VFL.

Table 5: Comparison of Different VFL Architectures

| Framework | Architecture | Asynchronous | Computational Efficiency | Comm. Mechanism | Scalability | Fault Tolerance | Implementation Complexity | Representative Frameworks |
|---|---|---|---|---|---|---|---|---|
| Pure VFL | Centralized | No | Low | Direct Peer-to-Peer | Low | Low | Low | N/A |
| VFL with PS | PS | No | High | Centralized PS Communication | Medium | Medium | Medium | FATE [22], PaddleFL [23] |
| AVFL | Centralized | Yes | Medium | Asynchronous Peer-to-Peer | Medium | Low | High | SecureBoost [50], AsyVFL |
| AVFL with PS | PS | Yes | High | Asynchronous PS Communication | High | Medium | High | Falcon [26] |
| PubSub-VFL | **Pub/Sub with PS** | **Yes** | **Highest** | **Efficient Pub/Sub Channels** | **Highest** | **High** | **Medium** | **Proposed System** |

# B  Publisher/Subscriber Architecture

**Pub/Sub Architecture.**  The Pub/Sub model consists of three primary entities: publishers, subscribers, and a message broker (or middleware). Specifically, the Pub/Sub architecture is a design pattern that facilitates communication between different parts of a software system by decoupling message senders (publishers) from message receivers (subscribers) [51]. In this model, publishers generate messages and send them to a central message broker, specifying topics or channels. Subscribers express interest in particular topics and receive messages related to those topics through the broker, without needing to know who the publishers are. This setup allows for scalable, flexible, and asynchronous communication, as publishers and subscribers can operate independently, and additional components can be added without disrupting existing interactions [52]. We summarize the advantages of the Pub/Sub architecture as follows:

- **Decoupling of Components:** Publishers and subscribers operate independently, reducing system complexity and enhancing maintainability.
- **Scalability:** The architecture supports high-throughput message dissemination, making it suitable for large-scale distributed systems.

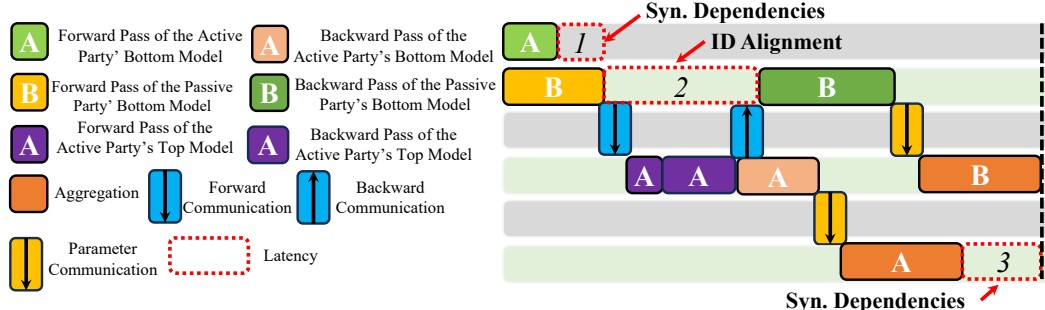

Figure 6: Pipeline overview of the VFL with PS architecture. Synchronization dependencies may cause latency 1 and latency 3, while latency 2 is the computation dependency caused by ID alignment.

- **Fault Tolerance:** Message brokers often provide mechanisms for persistence, ensuring reliability even in the presence of failures.
- **Asynchronous Communication:** Subscribers receive messages without blocking publishers, improving system responsiveness.

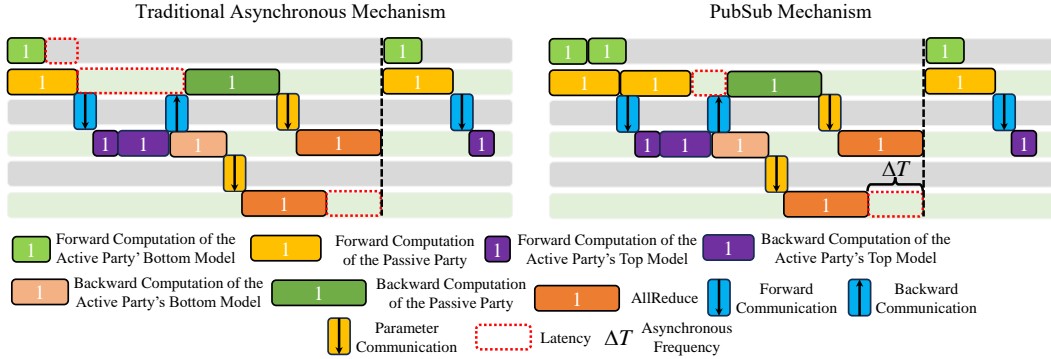

Figure 7: Traditional Asynchronous Mechanism *v.s.* Pub/Sub Mechanism.

**Traditional Asynchronous Mechanism *v.s.* Pub/Sub Mechanism.** In Fig. 7, we summarize the pipeline of the traditional asynchronous mechanism and the Pub/Sub mechanism in VFL. We found that the traditional asynchronous mechanism still has a lot of redundant latency because it is difficult to decouple from ID alignment. Because the traditional asynchronous mechanism relies on direct sender-receiver interactions using message queues, callbacks, or polling, leading to tight coupling and increased complexity as the number of workers grows. In contrast, the Pub/Sub mechanism introduces a decoupled communication model where publishers send messages to a broker, and subscribers receive relevant updates asynchronously, improving scalability and flexibility. Therefore, Pub/Sub can be well decoupled from ID alignment, so that the next task can be executed without additional waiting.

## C Differential Privacy Protocol

**Definition C.1** *A randomized mechanism $\mathcal{M}$ satisfies $(\mu, \sigma_{dp})$-GDP if for all measurable sets $S$ and for all adjacent datasets $D$ and $D'$, the following inequality holds:*

$$\mathbb{P}[\mathcal{M}(D) \in S] \leq e^{\mu}\mathbb{P}[\mathcal{M}(D') \in S] + \delta, \tag{16}$$

*where $\mu$ is a privacy loss parameter, $\sigma_{dp}$ is the standard deviation of the Gaussian noise added, and $\delta$ is a small probability allowing for a slight relaxation of the privacy guarantee. We explain the GDP design applicable to the embedding mechanism belows.*

Unlike traditional differentially private VFL [53], which applies DP noise directly to the gradient, in SL-based VFL, privacy protection must be applied to the embedding sent by $P_b$ to safeguard

intermediate interaction results. This necessity arises because adversaries commonly exploit embedding inversion attacks [49] to infer $P_b$'s private feature information. To ensure DP protection at the embedding level, we leverage the moments accountant technique introduced in [54] to precisely calibrate the DP noise. Specifically, we apply GDP by injecting randomized noise into selected neurons, effectively balancing privacy preservation and model utility while mitigating privacy risks in SL-based VFL. To this end, we set the variance of the Gaussian random neuron at the $l$-th layer as

$$\sigma_{dp} = \mathcal{O}(N_m \sqrt{K}/(\mu N)), \tag{17}$$

where $N_m$ is the size of minibatch used at worker, $N$ is the size of the whole batch, $K$ is the number of queries (i.e., the number of batches processed by $h$ at worker), then `PubSub-VFL` satisfies $\mu$-GDP for the data of worker. This method demonstrates the trade-off between accuracy and privacy. To increase privacy, i.e., decrease $\mu$ in (16), the variance of random neurons needs to be increased (cf. (17)). However, as the variance of random neurons increases, the variance of the stochastic gradient also increases, which will in turn lead to slower convergence.

**Discussion.** In practice, existing VFL frameworks like FATE often employ homomorphic encryption protocols to protect the privacy of intermediate results, such as embeddings or gradients. However, these cryptographic techniques come with significant computational overhead, leading to increased costs for participants. Given that the participants in our scenarios are institutions or enterprises with a low likelihood of engaging in malicious attacks, GDP emerges as a more economical and privacy-friendly solution. Furthermore, in our experiments, we thoroughly evaluate the effectiveness of GDP in defending against various advanced inference attacks. The results demonstrate that GDP offers robust protection, making it a practical choice for preserving privacy in VFL systems without compromising performance.

## D  Convergence Proof

### D.1  Assumptions

In this section, we provide the necessary assumptions as follows:

**Assumption D.1** *(Smoothness). If the global loss function $f(\theta)$ is $L$-smooth, thus, we have:*

$$\|\nabla f(\theta_1) - \nabla f(\theta_2)\| \leq L\|\theta_1 - \theta_2\|, \quad \forall \theta_1, \theta_2. \tag{18}$$

**Assumption D.2** *(Strong Convexity). For the purpose of this analysis, we assume that $f(\theta)$ is $\mu$-strongly convex:*

$$f(\theta_2) \geq f(\theta_1) + \langle \nabla f(\theta_1), \theta_2 - \theta_1 \rangle + \frac{\mu}{2}\|\theta_2 - \theta_1\|^2. \tag{19}$$

**Assumption D.3** *(Bounded Stochastic Gradient Variance). Let $g(\theta; \xi)$ denote the stochastic gradient computed on a mini-batch. Then we have:*

$$\mathbb{E}\left[\|g(\theta; \xi) - \nabla f(\theta)\|^2\right] \leq \sigma^2. \tag{20}$$

**Assumption D.4** *(Bounded Delay/Staleness). Due to semi-asynchronous updates, the gradient used at iteration $t$ is computed at a delayed parameter $\theta_{t-\tau(t)}$ with $\tau(t) \leq \Delta T$. We assume that the delay satisfies*

$$\|\nabla f(\theta_t) - \nabla f(\theta_{t-\tau(t)})\| \leq L \sum_{j=t-\tau(t)}^{t-1} \|\theta_{j+1} - \theta_j\|. \tag{21}$$

**Assumption D.5** *(Gaussian DP Noise Injection). The Gaussian noise $\xi_{dp}$ added to each exchanged embedding is independent with*

$$\xi_{dp} \sim \mathcal{N}(0, \sigma_{dp}^2 I), \tag{22}$$

*and the variance $\sigma_{dp}^2$ is determined by the calibration formula provided. Consequently, the effective update noise is increased to*

$$\sigma_{total}^2 = \sigma^2 + \sigma_{dp}^2. \tag{23}$$

**Assumption D.6** *(Reliable Communication under Capacity Constraints). The Pub/Sub mechanism guarantees that embeddings (of dimension $d$) are delivered within the delay bound $\tau$ provided that the channel capacity $p$ is not exceeded.*

## D.2 Convergence Analysis

Based on the necessary assumptions above, we write the following convergence objective.

**Theorem D.1** *If Assumptions D.1–D.6 are held and if the update rule*

$$\theta_{t+1} = \theta_t - \eta\Big(g_{t-\tau(t)} + \xi_{dp,t}\Big), \tag{24}$$

*is applied with a constant learning rate $\eta$ chosen sufficiently small (so that higher-order terms are negligible), there exist constants such that the expected optimality gap satisfies*

$$\mathbb{E}[f(\theta_t) - f(\theta^*)] \leq \left(1 - 2\mu\eta + O(\eta^2 L + \eta^3 L^2 \tau)\right)^t \left(f(\theta_0) - f(\theta^*)\right) + \frac{L\eta\left(\sigma^2 + \sigma_{dp}^2\right)}{4\mu}, \tag{25}$$

*where $\theta^*$ is the unique minimizer of $f(\theta)$.*

**Proof D.1** *First, we review the gradient update rule. The gradient update with delay and DP noise is written as:*

$$\theta_{t+1} = \theta_t - \eta\left(g_{t-\tau(t)} + \xi_{dp,t}\right), \tag{26}$$

*where $\xi_{dp,t}$ is the independent DP noise at iteration $t$. We then use the $L$-smoothness (refer to Assumption D.1), we have:*

$$f(\theta_{t+1}) \leq f(\theta_t) + \langle \nabla f(\theta_t), \theta_{t+1} - \theta_t \rangle + \frac{L}{2}\|\theta_{t+1} - \theta_t\|^2. \tag{27}$$

*We substitute the gradient update rule, thus, we have:*

$$f(\theta_{t+1}) \leq f(\theta_t) - \eta\langle \nabla f(\theta_t), g_{t-\tau(t)} + \xi_{dp,t}\rangle + \frac{L\eta^2}{2}\|g_{t-\tau(t)} + \xi_{dp,t}\|^2. \tag{28}$$

*Taking expectation conditioned on $\theta_t$ and noting that $\xi_{dp,t}$ is independent of $\theta_t$ with zero mean, we get:*

$$\begin{aligned} \mathbb{E}[f(\theta_{t+1})] \leq f(\theta_t) &- \eta\langle \nabla f(\theta_t), \mathbb{E}[g_{t-\tau(t)}]\rangle \\ &+ \frac{L\eta^2}{2}\mathbb{E}\left[\|g_{t-\tau(t)} + \xi_{dp,t}\|^2\right]. \end{aligned} \tag{29}$$

*Since $\mathbb{E}[g_{t-\tau(t)}] = \nabla f(\theta_{t-\tau(t)})$, and using the independence and zero mean of $\xi_{dp,t}$, thus, we have:*

$$\begin{aligned} \mathbb{E}\left[\|g_{t-\tau(t)} + \xi_{dp,t}\|^2\right] = &\|\nabla f(\theta_{t-\tau(t)})\|^2 \\ &+ \mathbb{E}\left[\|g_{t-\tau(t)} - \nabla f(\theta_{t-\tau(t)})\|^2\right] \\ &+ \mathbb{E}\left[\|\xi_{dp,t}\|^2\right]. \end{aligned} \tag{30}$$

*By assumptions Assumptions D.3 and D.6, this yields*

$$\mathbb{E}\left[\|g_{t-\tau(t)} + \xi_{dp,t}\|^2\right] \leq \|\nabla f(\theta_{t-\tau(t)})\|^2 + \sigma^2 + \sigma_{dp}^2 = \|\nabla f(\theta_{t-\tau(t)})\|^2 + \sigma_{total}^2. \tag{31}$$

*Thus, the expected loss satisfies:*

$$\begin{aligned} \mathbb{E}[f(\theta_{t+1})] \leq f(\theta_t) &- \eta\langle \nabla f(\theta_t), \nabla f(\theta_{t-\tau(t)})\rangle \\ &+ \frac{L\eta^2}{2}\left(\|\nabla f(\theta_{t-\tau(t)})\|^2 + \sigma_{total}^2\right). \end{aligned} \tag{32}$$

*Because of the delay, we need to relate $\nabla f(\theta_t)$ and $\nabla f(\theta_{t-\tau(t)})$. Under the smoothness assumption and bounded delay (refer to Assumption D.4), one can show (following standard asynchronous update arguments) that:*

$$\langle \nabla f(\theta_t), \nabla f(\theta_{t-\tau(t)})\rangle \geq \|\nabla f(\theta_t)\|^2 - \delta_t, \tag{33}$$

*with an error term $\delta_t$ that is on the order of $L\sum_{j=t-\tau(t)}^{t-1}\|\theta_{j+1} - \theta_j\|\,\|\nabla f(\theta_t)\|$. This error is typically bounded by a term proportional to $\eta^2 L^2 \tau \|\nabla f(\theta_t)\|^2$. Thus, for some constant $C_1$,*

$$\langle \nabla f(\theta_t), \nabla f(\theta_{t-\tau(t)})\rangle \geq \|\nabla f(\theta_t)\|^2 - C_1\eta^2 L^2 \tau \|\nabla f(\theta_t)\|^2. \tag{34}$$

*Similarly, we can upper bound $\|\nabla f(\theta_{t-\tau(t)})\|^2$ in terms of $\|\nabla f(\theta_t)\|^2$ plus a similar delay-dependent error. For simplicity in the derivation, assume that the delay-related errors yield an additional multiplicative factor of order $C_1\eta^2 L^2\tau$.*

*Strong convexity (refer to Assumption D.2) implies:*

$$\|\nabla f(\theta_t)\|^2 \geq 2\mu\big(f(\theta_t) - f(\theta^*)\big), \tag{35}$$

*where $\theta^*$ is the unique minimizer.*

*Substitute the bounds back into the expected loss difference:*

$$\begin{aligned}
\mathbb{E}[f(\theta_{t+1})] &\leq f(\theta_t) - \eta\Big(\|\nabla f(\theta_t)\|^2 - C_1\eta^2 L^2\tau\,\|\nabla f(\theta_t)\|^2\Big)\\
&\quad + \frac{L\eta^2}{2}\Big(\|\nabla f(\theta_t)\|^2 + \sigma_{total}^2\Big)\\
&= f(\theta_t) - \eta\|\nabla f(\theta_t)\|^2\Big(1 - C_1\eta^2 L^2\tau\Big)\\
&\quad + \frac{L\eta^2}{2}\|\nabla f(\theta_t)\|^2 + \frac{L\eta^2}{2}\sigma_{total}^2.
\end{aligned} \tag{36}$$

*Collecting the gradient terms:*

$$\mathbb{E}[f(\theta_{t+1})] \leq f(\theta_t) - \eta\|\nabla f(\theta_t)\|^2\left(1 - C_1\eta^2 L^2\tau - \frac{L\eta}{2}\right) + \frac{L\eta^2}{2}\sigma_{total}^2. \tag{37}$$

*Using the strong convexity lower bound $\|\nabla f(\theta_t)\|^2 \geq 2\mu(f(\theta_t) - f(\theta^*))$, we obtain:*

$$\begin{aligned}
\mathbb{E}\big[f(\theta_{t+1}) - f(\theta^*)\big] &\leq \left(1 - 2\mu\eta\left(1 - C_1\eta^2 L^2\tau - \frac{L\eta}{2}\right)\right)\big(f(\theta_t) - f(\theta^*)\big)\\
&\quad + \frac{L\eta^2}{2}\sigma_{total}^2.
\end{aligned} \tag{38}$$

*For sufficiently small $\eta$ (and ignoring higher-order terms), this recursion can be written approximately as:*

$$\mathbb{E}\big[f(\theta_{t+1}) - f(\theta^*)\big] \leq \big(1 - \eta\mu + C_1'\eta^2 L^2\tau\big)\big(f(\theta_t) - f(\theta^*)\big) + \frac{C_2\eta^2 L}{2}\sigma_{total}^2,$$

*where $C_1'$ and $C_2$ are constants that incorporate the delay and smoothness effects.*

*Unrolling the recursion yields:*

$$\mathbb{E}\big[f(\theta_t) - f(\theta^*)\big] \leq \big(1 - \eta\mu + C_1'\eta^2 L^2\tau\big)^t\big(f(\theta_0) - f(\theta^*)\big) + \frac{C_2\eta L\,\sigma_{total}^2}{2\mu}, \tag{39}$$

*or equivalently,*

$$\begin{aligned}
\mathbb{E}\big[f(\theta_t) - f(\theta^*)\big] &\leq \big(1 - \eta\mu + C_1'\eta^2 L^2\tau\big)^t\big(f(\theta_0) - f(\theta^*)\big)\\
&\quad + \frac{C_2\eta L\,(\sigma^2 + \sigma_{dp}^2)}{2\mu}.
\end{aligned} \tag{40}$$

*Here, the second term represents the error floor caused by the combined stochastic gradient noise and the additional DP noise.*

**Remark on DP Noise.** *The additional term $\sigma_{dp}^2$ in the variance $\sigma_{total}^2$ means that even if the algorithm converges in expectation, the asymptotic error floor is increased by the amount of DP noise injected. In practice, this is the trade-off between privacy (controlled by the noise multiplier and hence $\sigma_{dp}^2$) and accuracy (since the convergence error floor grows with $\sigma_{dp}^2$).*

# E   Algorithms

The pseudo code of our improved `PubSub-VFL` training process and the dynamic programming algorithm are described as follows.

---

**Algorithm 1** PubSub-VFL Training Framework

---

**Require:** Active party $P_a$ with dataset $D_1$, Passive party $P_b$ with dataset $D_2$
**Ensure:** Trained models $f_a$, $f_p$, and $g$
 1: Initialize models $f_a$, $f_p$, and $g$ with random weights $\theta_1, \theta_2$
 2: Establish embedding channels $\mathcal{C}_e$ and gradient channels $\mathcal{C}_g$ with FIFO buffers
 3: Set synchronization interval $\Delta T_t$ using Eq. (5)
 4: **for** each global epoch $t = 1$ to $T$ **do**
 5:     **Publisher Phase (Passive Party $P_b$):**
 6:     **for** each worker $w_p \in P_b$ **do**
 7:         Sample batch $B_j$ with IDs $\{ID_1, ..., ID_B\}$
 8:         Compute embeddings $z^p = f_p(x^p; \theta_2)$
 9:         Add noise $\xi_{dp} \sim \mathcal{N}(0, \sigma_{dp}^2 I)$ to $z^p$ (GDP protocol)
10:         Publish $(z^p, B_j)$ to embedding channel $\mathcal{C}_e[B_j]$
11:     **end for**
12:     **Subscriber Phase (Active Party $P_a$):**
13:     **for** each worker $w_a \in P_a$ in parallel **do**
14:         **if** $\mathcal{C}_e[B_j]$ not empty **then**
15:             Fetch $(z^p, B_j)$ from $\mathcal{C}_e[B_j]$
16:             Compute $z^a = f_a(x^a; \theta_1)$ and $\hat{y} = g(z^a, z^p)$
17:             Calculate loss $\mathcal{L}(\hat{y}, y)$ and gradients $\nabla_{\theta_1}\mathcal{L}, \nabla_{z^p}\mathcal{L}$
18:             Publish $\nabla_{z^p}\mathcal{L}$ to gradient channel $\mathcal{C}_g[B_j]$
19:         **else**
20:             Trigger waiting deadline mechanism (skip after $T_{ddl}$)
21:         **end if**
22:     **end for**
23:     **Backward Propagation:**
24:     **for** each worker $w_p \in P_b$ **do**
25:         Fetch $\nabla_{z^p}\mathcal{L}$ from $\mathcal{C}_g[B_j]$
26:         Compute $\nabla_{\theta_2}\mathcal{L}$ and update $\theta_2 \leftarrow \theta_2 - \eta\nabla_{\theta_2}\mathcal{L}$
27:         Push $\theta_2$ to $P_b$'s parameter server
28:     **end for**
29:     **Semi-Async Parameter Aggregation:**
30:     **if** $t \mod \Delta T_t == 0$ **then**
31:         Aggregate $\theta_1$ from $P_a$'s workers via PS
32:         Broadcast updated $\theta_1$ to all $P_a$ workers
33:     **end if**
34: **end for**

---

# F  Dataset Information

We provide details of the benchmark datasets used as follows:

*Energy (Appliances Energy Prediction):* The Energy dataset consists of 19,735 samples with 27 features. It is used for regression tasks and focuses on predicting the energy consumption of appliances based on environmental and meteorological variables. This dataset is commonly used in energy efficiency and smart home applications.

*Blog (Blog Feedback Prediction):* The Blog dataset contains 60,021 samples and 280 features. It is a regression dataset designed for predicting the number of comments on blog posts based on textual and metadata features. The dataset originates from online blog platforms and is widely used in social media analytics.

*Bank (Bank Marketing):* The Bank dataset has 40,787 samples with 48 features. It is a classification dataset used for predicting whether a client will subscribe to a term deposit based on demographic, economic, and campaign-related features. This dataset is widely used in financial and marketing analytics.

*Credit (Credit Card Default Prediction):* The Credit dataset consists of 30,000 samples with 23 features. It is a classification dataset designed for predicting whether a credit card user will default on

**Algorithm 2** Optimal Configuration via Dynamic Programming

---

**Require:** CPU cores $C_a$, $C_p$, candidate batch sizes $\mathcal{B}$, memory constraints $M_A$, $M_P$
**Ensure:** Optimal $(w_a^*, w_p^*, B^*)$

1: Compute $B_{\max} \leftarrow \min\left( \left( \frac{M_A - M_{A0}}{\rho_A} \right)^{1/\chi}, \left( \frac{M_P - M_{P0}}{\rho_P} \right)^{1/\chi} \right)$
2: Initialize DP table $dp[i][j][r] \leftarrow \infty$
3: **for** each batch size $B_r \in \mathcal{B}$ where $B_r \leq B_{\max}$ **do**
4:     **for** each $w_a \in \{P, ..., Q\}$ **do**
5:         **for** each $w_p \in \{M, ..., N\}$ **do**
6:             Calculate computation delays $T_A$, $T_P$ using Eq. (7)-(9)
7:             Calculate communication delay $T_{comm} \leftarrow \frac{E+G}{B_b}$
8:             Total delay $\mathcal{O}(w_a, w_p, B_r) \leftarrow \max(T_A, T_P) + T_{comm}$
9:             **if** $O(w_a, w_p, B_r) < dp[w_a][w_p][B_r]$ **then**
10:                Update $dp[w_a][w_p][B_r] \leftarrow O(w_a, w_p, B_r)$
11:             **end if**
12:         **end for**
13:     **end for**
14: **end for**
15: Return $(w_a^*, w_p^*, B^*) \leftarrow \arg \min dp[\cdot][\cdot][\cdot]$

---

their next payment. The dataset is sourced from financial institutions and is commonly used in risk assessment and credit scoring.

Table 6: Summary of Benchmark Datasets for `PubSub-VFL` Evaluation.

| Dataset | Samples | Features | Task Type | Domain |
|---------|---------|----------|-----------|--------|
| Energy | 19,735 | 27 | Regression | Energy Efficiency |
| Blog | 60,021 | 280 | Regression | Social Media |
| Bank | 40,787 | 48 | Classification | Finance/Marketing |
| Credit | 30,000 | 23 | Classification | Finance |

# G  Embedding Inversion Attacks

**Embedding Inversion Attacks (EIA).** Embedding inversion attacks aim to recover private feature representations from embeddings shared in the VFL framework. When the bottom model contains only the embedding layer, the attacker predicts the original feature by finding the nearest neighbor of each perturbed embedding in the embedding space [55]. For bottom models with additional layers, a more sophisticated optimization-based attack [49] is used. This method iteratively refines word selection vectors by minimizing the distance between the predicted feature's representations and the observed representations for each input sample. In this paper, we follow [49] to assume the adversary trains a neural network to directly map embeddings back to their original inputs. Futhermore, we assume that the adversary has access to a shadow dataset similar to the target $P_b$'s data.

Table 7: Accuracy comparison on benchmark datasets with large model.

| Dataset | Metric | VFL | VFL-PS | AVFL | AVFL-PS | Ours |
|---------|--------|-----|--------|------|---------|------|
| Energy | **RMSE** | 84.24 | 86.14 | 83.97 | 84.29 | **83.94** |
| Blog | **RMSE** | 23.18 | 23.07 | 22.97 | 23.15 | **22.14** |
| Bank | **AUC** | 94.97 | 94.74 | 95.02 | 95.06 | **96.97** |
| Credit | **AUC** | 83.42 | 85.44 | 84.23 | 82.27 | **86.07** |
| Synthetic | **AUC** | 92.74 | 92.67 | 91.54 | 92.21 | **94.17** |

# H  Additional Experiments

**Empirical Experiments.** To determine the constant $\lambda_a, \gamma_a, \lambda_p, \gamma_p, \lambda_a', \gamma_a', \varphi_a, \beta_a, \varphi_p, \beta_p, \beta_a', \varphi_a'$ in the delay model, we conduct empirical experiments. Specifically, we utilize a ten-layer MLP as the bottom model and a two-layer MLP as the top model. We set $B =$

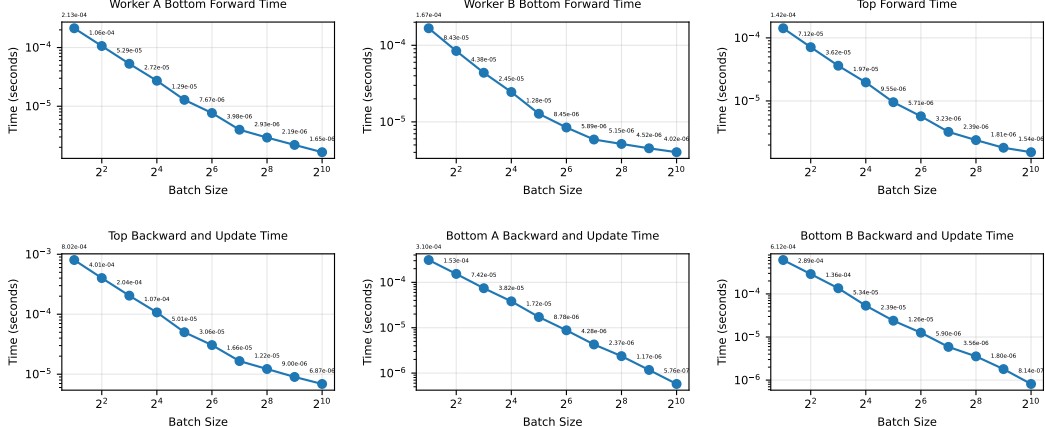

Figure 8: Empirical experimental results.

Table 8: The result of the constant being solved for.

| Symbol | Value | Symbol | Value | Symbol | Value |
|--------|-------|--------|-------|--------|-------|
| $\lambda_a$ | 0.018 | $\gamma_a$ | -0.8015 | $\lambda_p$ | 0.010 |
| $\gamma_p$ | -1.0071 | $\lambda_a'$ | 0.011 | $\gamma_p'$ | -0.7514 |
| $\varphi_a$ | 0.066 | $\beta_a$ | -0.6069 | $\varphi_p$ | 0.038 |
| $\beta_p$ | -1.0546 | $\beta_a'$ | -0.7834 | $\varphi_a'$ | 0.072 |

Table 9: Performance Comparison on Criteo 1TB Dataset.

| Dataset | Metric | VFL | VFL-PS | AVFL | AVFL-PS | Ours |
|---------|--------|-----|--------|------|---------|------|
| **Criteo 1TB** | AUC (%) | 81.23 | 81.45 | 80.97 | 81.32 | **82.15** |
| | Runtime (h) | 48.6 | 32.1 | 28.9 | 21.5 | **6.8** |
| | CPU Utilization (%) | 42.3 | 65.7 | 58.9 | 72.1 | **90.8** |
| | Waiting Time/epoch (s) | 12.8 | 8.5 | 6.2 | 4.1 | **1.3** |
| | Comm. Cost (GB) | 1280 | 950 | 890 | 720 | **450** |

$\{2, 4, 8, 16, 32, 64, 128, 256, 512, 1024\}$ to observe the forward and backward propagation times of both participants. The experimental results are presented in Fig. 8. Based on this figure and the delay model, we derive these constants, with their computed value shown in Table 8. Note that the constants solved in different operating environments are different.

**System Performance with Large Model.** We compare the performance of `PubSub-VFL` and its baselines on five datasets using large models. Similar to previous evaluations, we record the best performance results and configure these methods with their optimal hyperparameters. The experimental results are summarized in Table 7. The results show that both `PubSub-VFL` and its baselines remain unaffected by the large model (i.e., ResNet), with performance showing a slight improvement. This outcome demonstrates the robustness of `PubSub-VFL` in handling large models.

**System Performance on Large Dataset.** To further evaluate scalability, we introduce the Criteo 1TB Click Logs dataset[3], a widely used industrial benchmark for online advertising and recommendation systems. It contains ∼4.5 billion samples with 39 features (13 numerical, 26 categorical, and high-dimensional after one-hot encoding), representing real-world big data characteristics (massive samples and sparse features). Following the evaluation metrics (AUC for classification, runtime, CPU utilization, waiting time per epoch, and communication cost) in our paper, the Table 9 below compares `PubSub-VFL` with baselines. PubSub-VFL achieves superior performance in accuracy, efficiency, resource utilization, and communication cost. It attains the highest AUC of 82.15%, outperforming baselines by 0.7–1.2%, demonstrating strong robustness in large-scale, sparse data scenarios. With hierarchical asynchrony and Pub/Sub decoupling, it reduces runtime by ∼ 3× compared to AVFL-PS

---

[3]https://ailab.criteo.com/download-criteo-1tb-click-logs-dataset/

Table 10: Numerical results of system performance in a multi-party setting.

| Method (# of Parties) | Running Time (s) | CPU Utilization (%) | Waiting Time (s) | Comm. Cost (MB) | RMSE |
|---|---|---|---|---|---|
| PubSub-VFL (10) | 141.14 | 86.32 | 1.9273 | 896.34 | 23.44 |
| PubSub-VFL (8) | 121.55 | 88.36 | 2.0147 | 684.71 | 22.61 |
| PubSub-VFL (6) | 118.36 | 85.69 | 1.5697 | 645.34 | 22.34 |
| PubSub-VFL (4) | 104.72 | 90.14 | 1.2254 | 569.65 | 23.17 |
| PubSub-VFL (2) | 92.54 | 91.07 | 1.1389 | 439.45 | 22.34 |
| VFL-PS (10) | 1324.71 | 52.24 | 1.4410 | 1264.64 | 24.19 |
| VFL-PS (8) | 1374.63 | 47.64 | 1.2147 | 1165.17 | 22.61 |
| VFL-PS (6) | 1245.94 | 50.36 | 1.1647 | 1211.37 | 22.35 |
| VFL-PS (4) | 1174.65 | 51.24 | 1.4211 | 1089.64 | 23.19 |
| VFL-PS (2) | 974.65 | 41.47 | 1.2765 | 874.55 | 23.07 |
| AVFL (10) | 1445.28 | 27.65 | 20.3677 | 1024.34 | 23.54 |
| AVFL (8) | 1274.57 | 28.41 | 21.4154 | 967.57 | 23.71 |
| AVFL (6) | 1198.18 | 28.67 | 17.6517 | 915.16 | 24.01 |
| AVFL (4) | 1181.14 | 25.63 | 16.7456 | 847.65 | 22.84 |
| AVFL (2) | 1068.88 | 21.74 | 15.3657 | 754.77 | 22.97 |
| AVFL-PS (10) | 1274.51 | 67.51 | 2.6971 | 965.59 | 23.08 |
| AVFL-PS (8) | 1165.33 | 68.14 | 2.8146 | 817.55 | 23.67 |
| AVFL-PS (6) | 1017.82 | 58.59 | 2.6511 | 721.38 | 23.61 |
| AVFL-PS (4) | 1197.53 | 61.23 | 2.5636 | 617.45 | 24.07 |
| AVFL-PS (2) | 1057.67 | 57.68 | 2.4788 | 565.24 | 23.15 |

(6.8h vs. 21.5h) and $\sim 7\times$ compared to VFL (6.8h vs. 48.6h). It achieves high CPU utilization of 90.8%, indicating effective load balancing in heterogeneous environments, and cuts communication cost to 450GB—approximately 40% lower than AVFL-PS—through optimized channel management and reduced stale updates.

**System Performance in a Multi-party Setting.** While `PubSub-VFL` is currently designed and evaluated in a two-party VFL setting, its core architectural features suggest potential for extension to multi-party scenarios. The decoupled Publisher/Subscriber mechanism inherently supports many-to-many communication patterns, which is advantageous for scaling. Similarly, the hierarchical asynchronous design and buffering strategies can generalize to handle diverse update timings from multiple parties. However, the core challenges are the complexity of ID alignment and the adaptation of optimization algorithms. To address these two challenges, we provide the following two insights:

- **Complexity of ID Alignment.** To address this challenge, we can leverage existing multi-party PSI techniques (e.g., [36]), which can be applied during the system configuration phase.
- **Adaptation of Optimization Algorithms.** Because multiple parties are involved, the Dynamic Programming algorithm's search space becomes large, making it difficult to find the optimal solution. To address this challenge, a straightforward approach is to jointly model the passive party with the least resources (known from system profile information) and the active party to determine the optimal hyperparameter configuration. The key insight from doing so is that the key bottleneck dragging down system efficiency is the efficiency gap between the active party and the passive party with the least resources. This idea is consistent with the original manuscript. Although this approach is not optimal, it remains an option for expansion. In this way, our framework can be straightforwardly extended to multi-party scenarios.

The reason we did not include experiments in multi-party settings in the main text is due to the focused scope of the research topic and the intended application scenarios. We believe that incorporating both settings in a single paper might dilute the clarity and focus of the core contributions. To implement these improvements, we refactored the `PubSub-VFL` implementation by modifying several core components. Specifically, we improved the cache mechanism by increasing cache capacity to support more stable training, extended the wait deadline ($T_{ddl}$) to 15 seconds to enhance the reliability of embedding matching, and refined the dynamic programming optimization algorithm. All other system mechanisms were kept unchanged. Furthermore, we conducted a series of comparative experiments on the Blog dataset to evaluate the performance of PubSub-VFL in multi-party scenarios. The results are summarized in the Table 10.

