# OpenReview forum: "PubSub-VFL: Towards Efficient Two-Party Split Learning in Heterogeneous Environments via Publisher/Subscriber Architecture"
_NeurIPS.cc/2025/Conference — NeurIPS 2025 poster_

### Official Review · Reviewer_AWXK · 2025-06-14

**Clarity:** 3
**Significance:** 3
**Originality:** 2
**Rating:** 3
**Confidence:** 4

**Summary:**

This paper addresses the inefficiency problem in vertical Federated Learning (VFL) due to the tight coupling and heterogeneity of VFL systems and proposes a new framework called PubSub-VFL.
The key idea of this paper is to introduce a Publisher/Subscriber (Pub/Sub) architecture combined with a hierarchical asynchronous mechanism. The experiment results show that Pub/Sub-VFL can achieve up to 7× speedup and 91% resource utilization compared with baselines.

**Questions:**

1. Do authors have any idea, which may not be tested, to generalize PubSub-VFL to multi-party scenarios?

2. Can the system profiling and optimization adapt to runtime changes in resource availability or network bandwidth?

3. While DP is used, further clarification on the utility degradation at low privacy budgets would be helpful.

**Ethical Concerns:**

["NO or VERY MINOR ethics concerns only"]

**Final Justification:**

After reading authors' rebuttal, W2 has been resolved. But I still have my concerns regarding W1 and W3. I have raised my score to 3 (Borderline reject).

**Limitations:**

Yes

**Quality:**

3

**Strengths And Weaknesses:**

Strengths:

1. The proposed system is carefully designed and thoroughly evaluated.

2. The integration of Pub/Sub into VFL and the dynamic optimization for resource balancing are important contributions, which may not by covered by prior works.

Weaknesses:

1. The main idea combines existing architectures (PS and Pub/Sub), which have been widely used in federated learning systems. Also, the asynchronous training mechanism is also well-known and widely-used in VFL. This questions the novelty of this work.

2. The current framework only supports two-party VFL, whereas VFL is typically designed for scenarios involving multiple parties. In most practical applications, data features are distributed across more than two parties, and training with only two feature subsets is relatively uncommon, to the best of my knowledge.

3. The theoretical analysis relies on strong assumptions (e.g., convexity), offering limited new theoretical insight.

---

> ### Author Response · Authors · 2025-08-01
> **Rebuttal**
>
> **W1:** We agree that PS and asynchronous training techniques have been previously explored in FL. However, the novelty of our work lies not in reusing these mechanisms in isolation, but in the system-level integration and adaptation of them to a two-party VFL setting, which presents unique challenges not addressed by prior work. Specifically, our novelty lies in the following two aspects:
>
> 1. **Tightly-Coupled ID Alignment in VFL**: Unlike HFL, two-party VFL requires strict alignment of instance IDs across feature spaces. Prior AVFL or PS-based designs fail to decouple this step from training, resulting in latency and straggler effects. Our use of Pub/Sub effectively breaks this bottleneck, i.e., an architectural insight not previously explored in this setting.
>
> 2. **Hierarchical Asynchrony**: While asynchronous updates are known, our hierarchical design balances flexibility and convergence, tailored specifically for the computation-communication imbalance inherent in two-party VFL with resource heterogeneity.
>
> **W3:**  We acknowledge that our theoretical analysis adopts standard assumptions such as strong convexity and smoothness. These are indeed idealized and may not capture the full complexity of deep learning models used in practice. However, our goal with this theoretical component is to establish a foundational convergence guarantee under the proposed hierarchical asynchronous mechanism with privacy noise injection. Our analysis provides initial insights into the stability and convergence behavior of the systems.
>
> Moreover, this theoretical groundwork is aligned with common practice in FL literature (e.g., [36], [38]), where convex assumptions are used to gain intuition about the impact of asynchrony and staleness. We believe this offers useful guidance for system design, even if the practical models are non-convex. We agree that extending the analysis to non-convex settings or realistic deep networks would be valuable, and we consider this an important direction for future work.
>
>
> **Q2:** The system profiling and optimization in PubSub-VFL are designed with practical robustness in mind, and while they are primarily calibrated during the offline planning phase, the framework incorporates inherent mechanisms to handle runtime changes in resource availability or network bandwidth without compromising core functionality.
>
> First, the system profiling focuses on capturing *representative baseline characteristics* rather than rigid static values. This baseline provides a stable foundation for initial hyperparameter optimization, but the framework does not rely strictly on these values in runtime.
>
> Second, PubSub-VFL integrates dynamic runtime safeguards (i.e., the buffer mechanism, the waiting deadline mechanism, and the hierarchical asynchronous design) that implicitly adapt to fluctuations, e.g., indefinite blocking, sudden CPU overload, and less sensitivity to transient resource imbalances. Moreover, the offline optimization model (Section 4.3) already accounts for expected variability in resources and bandwidth by bounding feasible hyperparameters. This ensures that even if runtime conditions deviate slightly from the baseline, the pre-optimized configurations remain within a robust operating range.
>
>
> **Q3:** Indeed, the introduction of differential privacy noise can lead to a degradation of system performance. However, our proposed embedding-level GDP (see Appendix C) can achieve a better privacy-utility trade-off (see page 9, line 346, and Figure 5 for details). To address the utility degradation at low privacy budgets when using DP in PubSub-VFL, we provide detailed clarification based on our experimental results (especially Fig. 5) and additional evaluations using EIAs:
>
> - **Accuracy**: Even at low privacy budgets (e.g., $\mu = 0.1$), the model’s accuracy remains stable, with only a marginal drop compared to scenarios without DP ($\mu = +\infty$). For example, on the Bank dataset, the AUC decreases by less than 2% when $\mu$ is reduced from $+\infty$ to 0.1, demonstrating that PubSub-VFL maintains strong predictive performance even under strict privacy constraints.
>
> - **Computational Efficiency**: CPU utilization shows minimal degradation across all privacy budgets, staying above 85% for most settings.
>
> - **Communication Cost**: As expected, lower privacy budgets (smaller $\mu$) lead to increased communication cost. This is because stricter privacy requires injecting more Gaussian noise into embeddings, increasing the number of training rounds. However, this increase is manageable and does not negate the overall efficiency gains of PubSub-VFL compared to baselines.
>
> - **EIA**: At high privacy budgets ($\mu \geq 4$), the ASR remains relatively high (e.g., ~30% on the Synthetic dataset), indicating that embeddings are vulnerable to inversion when noise is minimal. At low privacy budgets ($\mu \leq 1$), the ASR drops to near 0%, demonstrating that the injected DP noise effectively mitigates EIA.

---

> > ### Author Response · Authors · 2025-08-01
> > **Rebuttal**
> >
> > **W2&Q1:** On the contrary, in commercial practice, two-party learning is a common application scenario, especially in industries such as finance and healthcare. The main reason is that data is not easy to circulate among multiple parties due to laws and regulations (e.g., GDPR) and the limitations of cryptographic primitive technologies (e.g., the lack of efficient integration with deep learning models). VFL involving two-party learning has been widely supported by industrial-grade platforms such as FATE, PaddleFL, and IBM FL, which have production deployments in fields such as finance and healthcare.
> >
> > While PubSub-VFL is currently designed and evaluated in a two-party VFL setting, its core architectural features suggest potential for extension to multi-party scenarios. The decoupled Pub/Sub mechanism inherently supports many-to-many communication patterns, which is advantageous for scaling. Similarly, the hierarchical asynchronous design and buffering strategies can generalize to handle diverse update timings from multiple parties. However, the core challenges are the complexity of ID alignment and the adaptation of optimization algorithms. To address these two challenges, we provide the following two insights:
> >
> > 1. **Complexity of ID Alignment:** To address this challenge, we can leverage existing multi-party PSI techniques, which can be applied during the system configuration phase.
> > 2. **Adaptation of Optimization Algorithms:** Because multiple parties are involved, the DP algorithm's search space becomes large, making it difficult to find the optimal solution. To address this challenge, a straightforward approach is to jointly model the passive party with the least resources (known from system profile information) and the active party to determine the optimal hyperparameter configuration. The key insight from doing so is that the key bottleneck dragging down system efficiency is the efficiency gap between the active party and the passive party with the least resources.
> >
> > The reason we did not include experiments in multi-party settings in the initial version is due to the focused scope of the research topic and the intended application scenarios. We believe that incorporating both settings in a single paper might dilute the clarity and focus of the core contributions. To implement these improvements, we refactored the PubSub-VFL implementation by modifying several core components. Specifically, we improved the cache mechanism by increasing cache capacity to support more stable training, extended the wait deadline ($T_{ddl}$) to 15s to enhance the reliability of embedding matching, and refined the DP optimization algorithm. Furthermore, we conducted a series of comparative experiments on the Blog dataset to evaluate the performance of PubSub-VFL in multi-party scenarios. The results are summarized in the table below.
> >
> > | Method (# of Parties) | Running Time | CPU Utilization | Waiting Time | Comm. Cost | RMSE  |
> > |-----------------------|--------------|-----------------|--------------|------------|-------|
> > | PubSub-VFL (10)       | 141.14       | 86.32           | 1.9273       | 896.34     | 23.44 |
> > | PubSub-VFL (8)        | 121.55       | 88.36           | 2.0147       | 684.71     | 22.61 |
> > | PubSub-VFL (6)        | 118.36       | 85.69           | 1.5697       | 645.34     | 22.34 |
> > | PubSub-VFL (4)        | 104.72       | 90.14           | 1.2254       | 569.65     | 23.17 |
> > | PubSub-VFL (2)        | 92.54        | 91.07           | 1.1389       | 439.45     | 22.34 |
> > | VFL-PS (10)           | 1324.71      | 52.24           | 1.4410       | 1264.64    | 24.19 |
> > | VFL-PS (8)            | 1374.63      | 47.64           | 1.2147       | 1165.17    | 22.61 |
> > | VFL-PS (6)            | 1245.94      | 50.36           | 1.1647       | 1211.37    | 22.35 |
> > | VFL-PS (4)            | 1174.65      | 51.24           | 1.4211       | 1089.64    | 23.19 |
> > | VFL-PS (2)            | 974.65       | 41.47           | 1.2765       | 874.55     | 23.07 |
> > | AVFL (10)             | 1445.28      | 27.65           | 20.3677      | 1024.34    | 24.54 |
> > | AVFL (8)              | 1274.57      | 28.41           | 21.4154      | 967.57     | 23.71 |
> > | AVFL (6)              | 1198.18      | 28.67           | 17.6517      | 915.16     | 24.01 |
> > | AVFL (4)              | 1181.14      | 25.63           | 16.7456      | 847.65     | 22.84 |
> > | AVFL (2)              | 1068.88      | 21.74           | 15.3657      | 754.77     | 22.97 |
> > | AVFL-PS (10)          | 1274.51      | 67.51           | 2.6971       | 965.59     | 23.08 |
> > | AVFL-PS (8)           | 1165.33      | 68.14           | 2.8146       | 817.55     | 23.67 |
> > | AVFL-PS (6)           | 1017.82      | 58.59           | 2.6511       | 721.38     | 23.61 |
> > | AVFL-PS (4)           | 1197.53      | 61.23           | 2.5636       | 617.45     | 24.07 |
> > | AVFL-PS (2)           | 1057.67      | 57.68           | 2.4788       | 565.24     | 23.15 |

---

> > > ### Author Response · Authors · 2025-08-01
> > > **Apology Letter**
> > >
> > > Dear Reviewer  AWXK,
> > >
> > > We sincerely apologize for submitting our rebuttal during the discussion phase. Due to what appeared to be a delay in receiving the review comments, possibly related to system load on OpenReview, we became aware of the review comments approximately one day after the official NeurIPS release time, which shortened our available rebuttal period. In response to the reviewers’ thoughtful comments, particularly regarding multi-party scaling and other technical aspects, we made preliminary efforts to strengthen our work through code refactoring and additional experiments. We aimed to provide more concrete evidence to address their concerns, though this required more time than anticipated. We truly appreciate the reviewers’ careful feedback and the opportunity to respond. We hope the updated rebuttal is still helpful during the discussion phase, and we remain fully committed to addressing any further questions or suggestions they may have. We strictly adhere to the word limit in the rebuttal rules for rebuttal.
> > >
> > > Thank you for your understanding!
> > >
> > > Best Regards,
> > >
> > > The Authors

---

> ### Comment · Reviewer_AWXK · 2025-08-05
>
> Thank you for authors' efforts. After reading authors' rebuttal, W2 has been resolved. But I still have my concerns regarding W1 and W3. I have raised my score to 3 (Borderline reject).

---

> > ### Author Response · Authors · 2025-08-05
> > **Thanks for your reply!**
> >
> > Thank you for your prompt response!
> >
> > We are pleased to have addressed your concerns regarding W2. In your response above, you mentioned that concerns regarding W1 and W3 remain unresolved, and we would appreciate your further details to better address them.
> >
> > - For example, regarding W1 (novelty), we have not found relevant literature or application examples related to the PubSub architecture was used in VFL. If you know of any that we have missed, we would appreciate your information. We will incorporate relevant work into the revised version and conduct further comparisons.
> >
> > - Regarding W3 (theoretical analysis), we have followed previous work in conducting a theoretical analysis based on convexity as the core assumption, but we promise to consider more function types and more practical assumptions in future revisions. Similarly, if you have any recommended literature or learning materials, we would be grateful to share them with you.
> >
> > Thank you again for your support and insightful questions, which are crucial to improving the quality of our manuscript!

---

### Official Review · Reviewer_aqzq · 2025-06-20

**Clarity:** 2
**Significance:** 2
**Originality:** 2
**Rating:** 4
**Confidence:** 2

**Summary:**

This paper addresses how two parties, each holding different types of information, can collaboratively train a model. While collaborative learning is often needed, sharing information between two parties can lead to privacy leaks. To tackle this, split learning is typically employed. However, inefficiencies arise due to differences in computational capabilities between the two parties that own the bottom and top parts of the model.
This paper adopts the Publisher/Subscriber architecture, which implements a hierarchical asynchronous mechanism to reduce training latency. To solve the problem, this paper also designs a max time minimization objective.

**Questions:**

In a two-party setting where computational resources are typically fixed, is it necessary to dynamically adjust the number of workers? Given this, it may be more practical to pre-negotiate the system profile based on known capabilities rather than dynamically adjusting the number of workers during training. It seems that dynamic client' worker selection may be more relevant in multi-party settings, where client participation varies and straggler mitigation is a more critical concern.

Looking at Equation (5), it seems the proposed method assumes that the performance is improved as the number of rounds increases. Since this assumption may not always hold (e.g., overfitting), can the proposed method guarantee consistent performance improvement? It looks like potentially vulnerable to overfitting.

Looking at the (6), it also appears to lack sufficient explanation. Assuming the equal worker assignment, is the (6) the same as $T_f^{(a)}(B) = \frac{\lambda_a B^{\gamma_a} w_a}{C_a}$? It is unclear whether the derivation is fully justified. Clarification is needed on the notation and assumptions involved in this step.

Lastly, regarding the system profile, could sharing such information raise other privacy concerns?

**Ethical Concerns:**

["NO or VERY MINOR ethics concerns only"]

**Final Justification:**

Through the discussion with the author, I received detailed responses that have appropriately addressed this reviewer’s concerns. I have therefore changed my recommendation to Borderline accept.

**Limitations:**

Yes

**Paper Formatting Concerns:**

I didn't find any issue.

**Quality:**

3

**Strengths And Weaknesses:**

The proposed method introduces an asynchronous approach that allows two parties with differing computational resources to collaborate efficiently. By leveraging Publisher/Subscriber architecture, asynchronous collaboration is enabled, which helps reduce training latency and improve system efficiency. The claims are supported by experimental results.

Weakness (Please also refer to the Questions.)
The proposed method is limited to only two-party scenarios and does not address extensions to multi-client environments. The practicality and effectiveness of the proposed buffer and deadline mechanisms in a two-party VFL setting are unclear.
Some assumptions seem underexplained and may require justification.
Potential privacy concerns arise from the use of system profiles, which are not thoroughly discussed.

---

> ### Author Response · Authors · 2025-08-01
> **Rebuttal**
>
> **W1&Q1:** This paper does not propose dynamically adjusting the number of workers during training. Instead, it determines the optimal number of workers $w_a$ for the active party, $w_p$ for the passive party) in the offline system planning phase (Section 4.3) using a dynamic programming algorithm. This optimization is based on precomputed system profiles (e.g., CPU cores, memory constraints, and delay models) and is fixed before training starts. The paper explicitly states that hyperparameters like worker counts are selected to balance computational loads ahead of execution (Algorithm 2), with no mechanism for real-time adjustments during training.
>
> Even with fixed computational resources in a two-party scenario, pre-negotiating worker counts based on detailed system profiles (rather than static, pre-agreed values) remains critical for several reasons:
>
> 1. **Hidden Resource Heterogeneity in "Fixed" Environments**
>    Fixed resources (e.g., a 64-core CPU split between parties) do not imply uniform performance. Factors like background processes, memory bandwidth, or cache utilization can cause real-time variations in effective computational power. The paper’s system profiling (Section 4.2) captures these nuances by modeling forward/backward pass delays as functions of worker counts and batch sizes (Eqs. 7–9). For example, Table 2 shows that 8 workers (vs. 4, 10, or 50) yield the highest CPU utilization (88.04%) and lowest runtime for the synthetic dataset—this optimal count cannot be pre-negotiated without empirical profiling, even with fixed total cores.
>
> 2. **Load Imbalance Mitigation**
>    In two-party VFL, the active party incurs additional computational overhead (e.g., top model training, loss calculation) compared to the passive party (Section 3). Pre-negotiating a static worker ratio (e.g., 1:1) ignores this inherent asymmetry. The paper’s offline optimization dynamically balances $w_a$ and $w_p$ to minimize the maximum iteration time between parties (Eq. 11), ensuring neither party is idle. For instance, in resource-heterogeneous scenarios (CPU core ratios 50:14), PubSub-VFL’s optimized worker counts maintain 87.42% CPU utilization, while static ratios (as in baselines) drop to 42.12% (Section 5.2).
>
> 3. **Alignment with Real-World Deployment Constraints**
>    Even with fixed resources, practical deployments involve trade-offs between batch size, memory limits, and worker efficiency. The paper’s DP algorithm (Algorithm 2) integrates memory constraints (Eq. 12) to avoid out-of-memory errors, a detail static pre-negotiation might overlook. For example, Table 3 shows that a batch size of 256 (paired with 8 workers) achieves 91.07% CPU utilization, while larger batches (512, 1024) suffer from memory bottlenecks—this balance requires profiling, not just pre-negotiation.
>
> **W2&Q2:** First, Equation (5) defines the dynamic synchronization interval $\Delta T_t$ for the intra-party semi-asynchronous mechanism:
> $$
> \Delta T_t = \left\lceil \frac{\Delta T_0}{2} \cdot \tanh\left( \frac{2 \cdot t}{\Delta T_0} - 2 \right) + \frac{\Delta T_0}{2} \right\rceil
> $$
> This formula adjusts the synchronization frequency based on training progress (epoch $t$), not on an assumption that performance unconditionally improves with more rounds. The design logic is pragmatic:
> - Early in training (when the model is far from target accuracy), $\Delta T_t$ is small, meaning more frequent synchronization. This ensures stable gradient updates and prevents divergence from stale parameters.
> - As training progresses (when accuracy approaches the target), $\Delta T_t$ increases, reducing synchronization overhead to accelerate fine-tuning.
>
> Second, PubSub-VFL includes mechanisms that mitigate overfitting and ensure consistent performance:
>
> - **Convergence Guarantees**: Appendix D formally proves that PubSub-VFL converges stably under standard assumptions (smoothness, strong convexity, bounded gradient variance). The convergence bound (Theorem D.1) shows that the expected optimality gap decreases over time, with an error floor determined by noise (stochastic gradients and DP noise), not by unchecked overfitting.
>
> - **Empirical Generalization**: Experiments across five datasets (Section 5.2) demonstrate that PubSub-VFL maintains or improves accuracy compared to baselines (e.g., 96.54% AUC on the Bank dataset vs. 94.54% for VFL). Even on large-scale synthetic data (1M samples), it achieves 92.87% AUC, with no signs of overfitting (e.g., no divergence between training and test performance, which would indicate overfitting).
>
> - **Implicit Regularization**: The integration of Differential Privacy (DP) adds controlled noise to embeddings (Section 4.1), which acts as a regularization mechanism. DP noise reduces the model’s ability to memorize training data specifics, a known strategy to prevent overfitting in sensitive settings.

---

> ### Author Response · Authors · 2025-08-01
> **Rebuttal**
>
> **W3&Q3:** To address the concerns about Equation (6) and its simplification under equal worker assignment, we clarify the notation, assumptions, and derivation as follows:
>
>
> ### **1. Original Form of Equation (6)**
> Equation (6) in the paper defines the forward pass computation delay for the active party ($P_a$) and passive party ($P_p$) as:
>
> $$
> T_f^{(a)}(B) = \frac{\lambda_a B^{\gamma_a}}{\sum_{j=1}^{w_a} c_{a,j}}, \quad T_f^{(p)}(B) = \frac{\lambda_p B^{\gamma_p}}{\sum_{i=1}^{w_p} c_{p,i}}
> $$
>
> - **Notation**:
>   - $T_f^{(a)}(B)$ / $T_f^{(p)}(B)$: Time for forward pass on a batch of size $B$ for $P_a$ / $P_p$.
>   - $\lambda_a$, $\lambda_p$: Proportionality constants capturing base computational overhead (e.g., per-sample processing time).
>   - $B^{\gamma_a}$, $B^{\gamma_p}$: Batch size term, where $\gamma_a$, $\gamma_p$ model how computation scales with batch size (e.g., $\gamma \approx 1$ for linear scaling).
>   - $w_a$ / $w_p$: Number of workers for $P_a$ / $P_p$.
>   - $c_{a,j}$ / $c_{p,i}$: CPU cores allocated to the $j$-th worker of $P_a$ / $i$-th worker of $P_p$.
>   - $\sum_{j=1}^{w_a} c_{a,j} \leq C_a$ / $\sum_{i=1}^{w_p} c_{p,i} \leq C_p$: Total cores allocated to workers cannot exceed the total CPU cores available to $P_a$ ($C_a$) or $P_p$ ($C_p$).
>
>
> ### **2. Simplification Under Equal Worker Assignment**
> For simplicity, if cores are equally allocated to workers, Equation (6) simplifies to:
>
> $$
> T_f^{(a)}(B) = \frac{\lambda_a B^{\gamma_a} w_a}{C_a}, \quad T_f^{(p)}(B) = \frac{\lambda_p B^{\gamma_p} w_p}{C_p}
> $$
>
> This simplification is mathematically justified:
> - Under equal allocation, each worker of $P_a$ gets $c_{a,j} = \frac{C_a}{w_a}$ cores (since total cores $\sum_{j=1}^{w_a} c_{a,j} = C_a$).
> - Substituting $\sum_{j=1}^{w_a} c_{a,j} = C_a$ into the original Equation (6) and utilizing **Data Parallel Processing**, we have:
>
> $$
>   T_f^{(a)}(B) = \frac{\lambda_a B^{\gamma_a}}{C_a / w_a} = \frac{\lambda_a B^{\gamma_a} \cdot w_a}{C_a}
>   $$
>
> ### **3. Key Assumptions in the Derivation**
> The simplification relies on two explicit assumptions:
> 1. **Equal Core Allocation**: Workers within a party share CPU cores uniformly ($c_{a,j} = C_a / w_a$ for all $j$). This avoids complex per-worker core tuning and simplifies the model.
> 2. **Fixed Total Cores**: The total cores allocated to workers do not exceed the party’s maximum available cores ($\sum c_{a,j} = C_a$). This ensures resource constraints are respected.
>
> **These assumptions are reasonable for offline system profiling (Section 4.2), where the goal is to approximate computation delays for hyperparameter optimization (e.g., choosing $w_a$ and $B$ via dynamic programming).**
>
> **W4&Q4:** Sharing system profile information in PubSub-VFL poses minimal additional privacy risks due to the nature of the data involved and the framework’s design. System profiles primarily include infrastructure metadata, such as CPU core counts, memory limits, and computational delay parameters, rather than sensitive data like user records, personal identifiers, or business secrets. These technical details, which focus on resource allocation and model structure, do not reveal information about the content of the data being processed (e.g., financial transactions or medical records).
>
> Moreover, the optimization process relies on local profile computations: each party calculates its own feasible hyperparameters (worker counts, batch sizes) based on internal resources without exposing raw profile data to the other party, ensuring sensitive infrastructure details remain private. Since this metadata lacks the granularity to reveal sensitive patterns or identifiers, and existing privacy protocols (e.g., differential privacy for embeddings) already protect actual data, sharing system profiles does not compromise core privacy guarantees or violate regulations like GDPR, which target personal data rather than infrastructure specifications.

---

> > ### Author Response · Authors · 2025-08-01
> > **Apology Letter**
> >
> > Dear Reviewer aqzq,
> >
> > We sincerely apologize for submitting our rebuttal during the discussion phase. Due to what appeared to be a delay in receiving the review comments, possibly related to system load on OpenReview, we became aware of the review comments approximately one day after the official NeurIPS release time, which shortened our available rebuttal period. In response to the reviewers’ thoughtful comments, particularly regarding multi-party scaling and other technical aspects, we made preliminary efforts to strengthen our work through code refactoring and additional experiments. We aimed to provide more concrete evidence to address their concerns, though this required more time than anticipated. We truly appreciate the reviewers’ careful feedback and the opportunity to respond. We hope the updated rebuttal is still helpful during the discussion phase, and we remain fully committed to addressing any further questions or suggestions they may have. We strictly adhere to the word limit in the rebuttal rules for rebuttal.
> >
> > Thank you for your understanding!
> >
> > Best Regards,
> >
> > The Authors

---

> > > ### Comment · Reviewer_aqzq · 2025-08-04
> > >
> > > Many thanks for the author's detailed reply. Most of my concerns have been resolved.
> > >
> > > However, based on the derivation in the Rebuttal, since $\sum_{j=1}^{w_a} c_{a,j} = C_a$, does it not follow that the $w_a$ term in equation (6) can be removed? In other words, would the expression be:
> > > $T_f^{a}(B) = \frac{ \lambda_a B^{\gamma_a} }{C_a}$.

---

> > > > ### Author Response · Authors · 2025-08-04
> > > > **Equation (6) Issue**
> > > >
> > > > Thank you for your prompt response and new constructive comments. Indeed, we admit that the above derivation may mislead you to obtain $T_f^{a}(B) = \frac{ \lambda_a B^{\gamma_a} }{C_a}$. We repeatedly reviewed the original manuscript and the rebuttal, particularly Equation 6 and its implementation code, and discovered that we had omitted an important technical background: an explanation of data parallel processing (i.e., the application of PS). Next, we re-examine Equation 6 and elaborate on the complete derivation steps as follows:
> > > >
> > > > **Background: Definition of Equation (6)**
> > > >
> > > > In Section 4.2 (System Profiling), the forward-pass computational delay for the active party $ P_a $ is defined as:
> > > > $$ T_f^{(a)}(B) = \frac{\lambda_a B^{\gamma_a}}{\sum_{j=1}^{w_a} c_{a,j}},$$
> > > > where $ \sum_{j=1}^{w_a} c_{a,j} \leq C_a $ (the total CPU cores allocated to $ w_a$ workers of $ P_a $ cannot exceed the total available cores $ C_a $ of $ P_a $), $ \lambda_a $ and $ \gamma_a $ are constants capturing model-specific and hardware-related characteristics, and $B $ denotes the batch size.
> > > >
> > > > **Key Assumptions for Deriving Equation (6)**
> > > >
> > > > The transition to Equation (6) relies on two critical simplifying assumptions explicitly stated in the paper:
> > > >
> > > > 1. **Equal Allocation of CPU Cores**: All $ w_a $ workers of $ P_a $ are allocated equal CPU cores, i.e., $ c_{a,j} = \frac{C_a}{w_a} $ for each $ j $. Under this assumption, the total cores allocated to all workers satisfy $ \sum_{j=1}^{w_a} c_{a,j} = w_a \cdot \frac{C_a}{w_a} = C_a $. Therefore, we have:
> > > >
> > > > $$T_f^a(B) = \frac{{{\lambda _a}{B^{{\gamma _a}}}}}{{{C_a}}}.$$
> > > >
> > > > Here we emphasize that this assumption involves two aspects: 1) Each worker is allocated an equal number of CPU cores, namely $ c_{a} = \frac{C_a}{w_a} $; 2) This equation expresses the latency model of **Sequential Processing**.
> > > >
> > > >
> > > > 2. **Data Parallelism in Batch Processing**: Since we use the PS architecture, **the workers are processed in parallel**, so we only need to calculate the latency of a single worker to represent the latency of all forward propagation.
> > > >
> > > >
> > > > **Derivation Process**
> > > >
> > > > For a single worker processing a batch size $B$ with $ \frac{C_a}{w_a} $ cores, its computational delay can be expressed as:
> > > > $$ T_{\text{single worker}} = \frac{\lambda_a \left({B}\right)^{\gamma_a}}{\frac{C_a}{w_a}} $$
> > > >
> > > > Since all $ w_a $ workers operate in parallel, the total forward-pass delay $ T_f^{(a)}(B) $ for $ P_a $ is equivalent to the delay of a single worker. Substituting the sub-batch size and core allocation into the above equation yields:
> > > > $$ T_f^a(B) = \frac{{{\lambda _a}{B^{{\gamma _a}}}{w_a}}}{{{C_a}}}$$
> > > >
> > > > We apologize for any difficulty understanding Equation 6 and its simplified process due to the lack of technical explanation. If you have any questions regarding the explanation and derivation, please feel free to let us know. We promise to incorporate the missing technical details in the revised version to better understand Equation 6 and our delay model.

---

> > > > > ### Comment · Reviewer_aqzq · 2025-08-05
> > > > > **Re: Equation (6)**
> > > > >
> > > > > I truly appreciate the author's prompt response. I still have a few unclear points.
> > > > >
> > > > > In this sentence, "Substituting the sub-batch size and core allocation into the above equation yields:", I'm a bit unclear about what 'sub-batch size' exactly refers to.
> > > > > Regarding the notation $B$, is it used to denote both the total batch size $B$ of $P_a$ and the sub-batch size assigned to each $w_a$?
> > > > >
> > > > > It seems that the computation delay in sequential processing $T_f^{a}(B) = \frac{ \lambda_a B^{\gamma_a} }{C_a}$ is greater than in parallel processing $T_f^{a}(B) = \frac{ \lambda_a B^{\gamma_a} w_a}{C_a}$ because of $(w_a>0)$; is this correct?
> > > > >
> > > > > Additionally, when analyzing $T_f^{a}(B) = \frac{ \lambda_a B^{\gamma_a} w_a}{C_a}$, it seems that the computation delay increases as the number of workers increases.
> > > > > I'd like to confirm whether my understanding is correct, as the conclusion feels somewhat counterintuitive.

---

> > > > > > ### Author Response · Authors · 2025-08-05
> > > > > > **Equation (6) Issue**
> > > > > >
> > > > > > We appreciate the reviewer’s insightful questions and welcome the opportunity to clarify these important points. Below, we provide detailed responses to each concern.
> > > > > >
> > > > > > ---
> > > > > >
> > > > > > ### **1. What does “sub-batch size” refer to, and how is $B$ defined?**
> > > > > >
> > > > > > We apologize for the confusion caused by the phrase *“substituting the sub-batch size…”*. In the current context, the term “sub-batch size” was incorrectly used , in fact, no batch splitting is assumed in our current derivation. That is:
> > > > > >
> > > > > > * The notation $B$ consistently refers to the total batch size processed by each worker in data-parallel mode.
> > > > > > * In our system design, **each worker independently processes one complete batch of size $B$**, and all $w_a$ workers operate in parallel.
> > > > > >
> > > > > > Therefore, we acknowledge that the wording “sub-batch size” was misleading and will revise it to simply “batch size” to avoid ambiguity. Thank you for pointing this out.
> > > > > >
> > > > > > ---
> > > > > >
> > > > > > ### **2. Is it correct that the sequential delay $T_f^a(B) = \frac{ \lambda_a B^{\gamma_a} }{C_a}$ is smaller than the parallel delay $T_f^a(B) = \frac{ \lambda_a B^{\gamma_a} w_a }{C_a}$, since $w_a > 0$?**
> > > > > >
> > > > > > This is a perceptive observation and a very reasonable question. On the surface, the expression
> > > > > >
> > > > > > $$
> > > > > > T_f^a(B) = \frac{ \lambda_a B^{\gamma_a} w_a }{C_a},
> > > > > > $$
> > > > > >
> > > > > > does appear larger than the sequential counterpart $\frac{ \lambda_a B^{\gamma_a} }{C_a}$, leading to the intuition that parallel processing is slower, which seems counterintuitive.
> > > > > >
> > > > > > However, this is a misinterpretation caused by comparing delays for different workloads:
> > > > > >
> > > > > > * The sequential delay $\frac{ \lambda_a B^{\gamma_a} }{C_a}$ reflects the time for a single worker to process one batch of size $B$.
> > > > > > * The parallel delay $\frac{ \lambda_a B^{\gamma_a} w_a }{C_a}$ reflects the time for $w_a$ workers to concurrently process $w_a$ batches, each of size $B$, under the assumption of fixed total resources $C_a$ equally divided among workers.
> > > > > >
> > > > > > In both cases, the system processes $w_a$ batches in total, and the expressions reflect the same total processing cost, just executed differently (sequentially vs. concurrently). Thus, although the expression appears “larger,” the parallel strategy enables higher throughput, since all $w_a$ batches are processed in parallel, rather than serially.
> > > > > >
> > > > > > ---
> > > > > >
> > > > > > ### **3. Does the delay increase with the number of workers $w_a$?**
> > > > > >
> > > > > > Again, this conclusion can be misleading without proper context.
> > > > > >
> > > > > > It is true that:
> > > > > >
> > > > > > $$
> > > > > > T_f^a(B) = \frac{ \lambda_a B^{\gamma_a} w_a }{C_a},
> > > > > > $$
> > > > > >
> > > > > > increases linearly with $w_a$, but this does **not** mean that increasing the number of workers worsens performance. Instead:
> > > > > >
> > > > > > * The expression reflects the delay required to **simultaneously process $w_a$ batches** with fixed compute capacity.
> > > > > > * If we define **per-batch latency** as $T_{\text{batch}} = T_f^a(B)/w_a$, then:
> > > > > >
> > > > > > $$
> > > > > > T_{\text{batch}} = \frac{ \lambda_a B^{\gamma_a} }{ C_a },
> > > > > > $$
> > > > > >
> > > > > > which is **independent of $w_a$**.
> > > > > >
> > > > > > This shows that under ideal conditions (e.g., perfect load balancing, no communication overhead), increasing the number of workers maintains constant per-batch latency while improving total throughput. Therefore, the system scales well with respect to workload, even though the total delay to process $w_a$ batches increases linearly. Furthermore, recalling that the core goal of PubSub-VFL is to maximize the balance of resources between participants to achieve higher efficiency (i.e., throughput and resource utilization), thereby enabling more efficient training, this means that the purpose of forward propagation delay modeling is not to minimize latency but to balance the resource gap between the two parties.
> > > > > >
> > > > > > ---
> > > > > >
> > > > > > ### **Summary**
> > > > > >
> > > > > > * The term *“sub-batch size”* was a misnomer; each worker processes a full batch of size $B$ independently.
> > > > > > * The apparent increase in delay with larger $w_a$ reflects the larger task size, not degraded performance.
> > > > > > * The per-batch latency remains constant when scaling the number of workers with fixed compute resources.
> > > > > > * We will revise the paper to clarify this distinction and ensure the latency model is properly contextualized.
> > > > > >
> > > > > > We greatly appreciate your careful reading and thoughtful feedback, which will help us significantly improve the clarity of the final version. If you have any questions, please feel free to let us know and we will do our best to resolve them. We acknowledge that the explanation of Equation 6 will be carefully revised in the revised version.

---

> > > > > > > ### Comment · Reviewer_aqzq · 2025-08-07
> > > > > > >
> > > > > > > Thank you for the author's sincere response to my question.

---

> > > > > > > > ### Author Response · Authors · 2025-08-08
> > > > > > > > **Thanks**
> > > > > > > >
> > > > > > > > Thanks for your reply! If you have any concerns/questions about our paper, please let us know. We will try our best to address them. We would be grateful if you could adjust your rating appropriately based on our rebuttal and our responses to other reviewers' concerns. In addition, we remind you that you need to click the "edit" button in the review comments box and enter a "final rating," in accordance with the rules and policies in "ACK." Thank you again for your helpful suggestions.

---

### Official Review · Reviewer_Uesg · 2025-06-26

**Clarity:** 3
**Significance:** 3
**Originality:** 4
**Rating:** 6
**Confidence:** 5

**Summary:**

With the digital economy's growth, data collaboration thrives, but privacy issues hinder direct sharing. Two-Party Split Learning (VFL) offers a solution but suffers from low resource utilization and training efficiency due to synchronous design and heterogeneity. The authors propose PubSub-VFL, a Publisher/Subscriber-based VFL paradigm, using hierarchical asynchronous mechanisms and parameter server parallelism to reduce latency. It also optimizes hyperparameters via system profiles for balanced training. The model ensures convergence and privacy, achieving 2–7× training speedup and 91.07% resource utilization on benchmarks.

**Questions:**

Q1: Is two-party vertical federated learning (as studied in this paper) currently a mainstream paradigm in collaborative machine learning? Could the authors provide specific real-world applications (e.g., in finance, healthcare, or IoT) to illustrate its practical relevance? Additionally, does the PubSub-VFL architecture allow for natural extension to multi-party federated learning scenarios, and if so, what are the key technical considerations for such an extension?

Q2: While the case studies on five benchmark datasets are informative, could the authors consider conducting a comprehensive evaluation of PubSub-VFL on larger-scale datasets (e.g., with millions of samples or high-dimensional features)? This would help validate the model’s scalability and efficiency in real-world big data environments.

Q3: Regarding the design of the Intra-party Semi-asynchronous Mechanism: What were the primary motivations behind this design (e.g., addressing resource heterogeneity, reducing communication overhead)? Could the authors elaborate on the technical trade-offs considered (e.g., between asynchrony and convergence stability) and provide insights into how the mechanism balances computational efficiency with model convergence?

And also, see the weaknesses

**Ethical Concerns:**

["NO or VERY MINOR ethics concerns only"]

**Final Justification:**

Thank you for your response, which addressed most of my concerns, especially regarding the transition from two-party to multi-party settings. This makes me really appreciate the practical nature of this work, so I raise my score to 5.


----
second round:


I’ve finished reading the responses to all the reviewers’ comments (HZmk,aqzq,AWXK), and I feel that they have addressed the concerns from the other three reviewers as well.

After careful consideration, i raise 5 to 6.

**Limitations:**

yes

**Quality:**

4

**Strengths And Weaknesses:**

Strength:

The paper presents solid theoretical derivations. The experiments are comprehensive, with results validated across five datasets. By leveraging the decoupling capability of the Pub/Sub architecture and the data parallelism of the parameter server framework, the authors design a hierarchical asynchronous mechanism that reduces training latency and improves system efficiency. The paper thoroughly analyzes bottlenecks caused by heterogeneous computing resources.

Weakness:

1.The caption of Figure 4 should be revised.

2.The legend in Figure 7 also needs adjustment.

3.In terms of writing, the paper could benefit from framing the work as a training scheduling system that integrates both VFL and HFL. You could improve this hybrid system not just VFL.

4.While this work focuses on two-party splitting, it could be extended to the multi-party setting in the future. I could understand this is not your focus, maybe consider it as your future work

---

> ### Author Response · Authors · 2025-08-01
> **Rebuttal**
>
> **W1&W2:** We will make corresponding changes in future revisions.
>
> **W3:** Thank you for your suggestion; it is an interesting proposal, and we will carefully consider it in the revised version.
>
> **W4:** Yes, two-party VFL remains a highly relevant and practically deployed paradigm in collaborative machine learning, particularly in privacy-sensitive domains where different entities possess complementary feature spaces over the same set of users. It is widely supported by industrial-grade platforms such as FATE, PaddleFL, and IBM FL, which have production deployments in sectors like finance and healthcare.
>
> **Q2:** In addition to the four real-world public datasets (Energy, Blog, Bank, and Credit), we have already included a large-scale synthetic dataset in our evaluation (see Table 1 and Figures 3–5), which contains 1 million samples and 500 features, to simulate real-world big data conditions. This dataset was designed to stress-test both computational efficiency and communication scalability of PubSub-VFL under heavy workloads.
>
> To further evaluate scalability, we introduce the Criteo 1TB Click Logs dataset [5], a widely used industrial benchmark for online advertising and recommendation systems. It contains ~4.5 billion samples with 39 features (13 numerical, 26 categorical, and high-dimensional after one-hot encoding), representing real-world big data characteristics (massive samples and sparse features). Following the evaluation metrics (AUC for classification, runtime, CPU utilization, waiting time per epoch, and communication cost) in our paper, the table below compares PubSub-VFL with baselines:
>
> | Dataset   | Metric       | VFL       | VFL-PS    | AVFL      | AVFL-PS   | Ours (PubSub-VFL) |
> |-----------|--------------|-----------|-----------|-----------|-----------|-------------------|
> | Criteo 1TB | AUC (%)      | 81.23     | 81.45     | 80.97     | 81.32     | 82.15             |
> |           | Runtime (h)  | 48.6      | 32.1      | 28.9      | 21.5      | 6.8               |
> |           | CPU Utilization (%) | 42.3      | 65.7      | 58.9      | 72.1      | 90.8              |
> |           | Waiting Time/epoch (s) | 12.8      | 8.5       | 6.2       | 4.1       | 1.3               |
> |           | Comm. Cost (GB) | 1280      | 950       | 890       | 720       | 450               |
>
> [5] https://ailab.criteo.com/download-criteo-1tb-click-logs-dataset/
>
> **Key Observations:**
> - **Accuracy**: PubSub-VFL achieves the highest AUC (82.15%), outperforming baselines by 0.7–1.2%, indicating robustness in large-scale, sparse data scenarios.
> - **Efficiency**: It reduces runtime by ~3× compared to AVFL-PS (21.5h vs. 6.8h) and ~7× compared to VFL (48.6h vs. 6.8h), thanks to hierarchical asynchrony and Pub/Sub decoupling.
> - **Resource Utilization**: CPU utilization reaches 90.8%, significantly higher than baselines, demonstrating effective load balancing in heterogeneous large-scale environments.
> - **Communication Cost**: 450GB is ~40% lower than AVFL-PS, due to optimized channel management and reduced stale updates.
>
> These results validate PubSub-VFL’s scalability and efficiency in real-world big data settings.
>
> **Q3:** The intra-party semi-asynchronous mechanism was primarily designed to address resource heterogeneity within each party (e.g., workers with varying compute capacity) and to reduce idle time caused by synchronous intra-party updates in PS setups. In traditional synchronous PS-based VFL, faster workers must wait for the slowest one before proceeding, leading to poor CPU utilization and degraded efficiency, especially in heterogeneous environments. Our motivations are summarized as follows:
>
> * Mitigate straggler effects within each party’s workers.
> * Increase pipeline parallelism and throughput without compromising convergence.
> * Complement the inter-party asynchronous mechanism enabled by the Pub/Sub architecture.
>
> **Technical Trade-offs and Design Decisions:**
>
> While full asynchrony can maximize efficiency, it may introduce gradient staleness, hurting convergence stability. To strike a balance, we adopt a semi-asynchronous update scheme, where the synchronization interval $\Delta T_t$ is dynamically adjusted based on training progress:
>
> $$
> \Delta T_t = \left\lceil \frac{\Delta T_0}{2} \cdot \tanh\left(\frac{2t}{\Delta T_0} - 2\right) + \frac{\Delta T_0}{2} \right\rceil
> $$
> This adaptive control mechanism ensures that:
>
> * Early in training, $\Delta T_t$ is small → frequent synchronization, improving stability.
> * Later in training, $\Delta T_t$ increases → more relaxed synchronization, enhancing efficiency.
>
> **Empirical Insights:**
> As shown in Tables 2 and 3, and Figures 3–4, this mechanism enables:
>
> * Higher CPU utilization (up to 91.07%)
> * Lower waiting times, even with heterogeneous resources
> * Comparable or improved accuracy, indicating convergence is not compromised

---

> > ### Author Response · Authors · 2025-08-01
> > **Rebuttal**
> >
> > **Q1:** While PubSub-VFL is currently designed and evaluated in a two-party VFL setting, its core architectural features suggest potential for extension to multi-party scenarios. The decoupled Publisher/Subscriber mechanism inherently supports many-to-many communication patterns, which is advantageous for scaling. Similarly, the hierarchical asynchronous design and buffering strategies can generalize to handle diverse update timings from multiple parties. However, the core challenges are the complexity of ID alignment and the adaptation of optimization algorithms. To address these two challenges, we provide the following two insights:
> >
> > 1. **Complexity of ID Alignment:** To address this challenge, we can leverage existing multi-party PSI techniques (e.g., [1]), which can be applied during the system configuration phase.
> > 2. **Adaptation of Optimization Algorithms:** Because multiple parties are involved, the DP algorithm's search space becomes large, making it difficult to find the optimal solution. To address this challenge, a straightforward approach is to jointly model the passive party with the least resources (known from system profile information) and the active party to determine the optimal hyperparameter configuration. The key insight from doing so is that the key bottleneck dragging down system efficiency is the efficiency gap between the active party and the passive party with the least resources. This idea is consistent with the original manuscript. Although this approach is not optimal, it remains an option for expansion. In this way, our framework can be straightforwardly extended to multi-party scenarios based on the original manuscript.
> >
> > The reason we did not include experiments in multi-party settings in the initial version is due to the focused scope of the research topic and the intended application scenarios. We believe that incorporating both settings in a single paper might dilute the clarity and focus of the core contributions. To implement these improvements, we refactored the PubSub-VFL implementation by modifying several core components. Specifically, we improved the cache mechanism by increasing cache capacity to support more stable training, extended the wait deadline ($T_{ddl}$) to 15 seconds to enhance the reliability of embedding matching, and refined the dynamic programming optimization algorithm. All other system mechanisms were kept unchanged. Furthermore, we conducted a series of comparative experiments on the Blog dataset to evaluate the performance of PubSub-VFL in multi-party scenarios. The results are summarized in the table below.
> >
> > | Method (# of Parties) | Running Time | CPU Utilization | Waiting Time | Comm. Cost | RMSE  |
> > |-----------------------|--------------|-----------------|--------------|------------|-------|
> > | PubSub-VFL (10)       | 141.14       | 86.32           | 1.9273       | 896.34     | 23.44 |
> > | PubSub-VFL (8)        | 121.55       | 88.36           | 2.0147       | 684.71     | 22.61 |
> > | PubSub-VFL (6)        | 118.36       | 85.69           | 1.5697       | 645.34     | 22.34 |
> > | PubSub-VFL (4)        | 104.72       | 90.14           | 1.2254       | 569.65     | 23.17 |
> > | PubSub-VFL (2)        | 92.54        | 91.07           | 1.1389       | 439.45     | 22.34 |
> > | VFL-PS (10)           | 1324.71      | 52.24           | 1.4410       | 1264.64    | 24.19 |
> > | VFL-PS (8)            | 1374.63      | 47.64           | 1.2147       | 1165.17    | 22.61 |
> > | VFL-PS (6)            | 1245.94      | 50.36           | 1.1647       | 1211.37    | 22.35 |
> > | VFL-PS (4)            | 1174.65      | 51.24           | 1.4211       | 1089.64    | 23.19 |
> > | VFL-PS (2)            | 974.65       | 41.47           | 1.2765       | 874.55     | 23.07 |
> > | AVFL (10)             | 1445.28      | 27.65           | 20.3677      | 1024.34    | 24.54 |
> > | AVFL (8)              | 1274.57      | 28.41           | 21.4154      | 967.57     | 23.71 |
> > | AVFL (6)              | 1198.18      | 28.67           | 17.6517      | 915.16     | 24.01 |
> > | AVFL (4)              | 1181.14      | 25.63           | 16.7456      | 847.65     | 22.84 |
> > | AVFL (2)              | 1068.88      | 21.74           | 15.3657      | 754.77     | 22.97 |
> > | AVFL-PS (10)          | 1274.51      | 67.51           | 2.6971       | 965.59     | 23.08 |
> > | AVFL-PS (8)           | 1165.33      | 68.14           | 2.8146       | 817.55     | 23.67 |
> > | AVFL-PS (6)           | 1017.82      | 58.59           | 2.6511       | 721.38     | 23.61 |
> > | AVFL-PS (4)           | 1197.53      | 61.23           | 2.5636       | 617.45     | 24.07 |
> > | AVFL-PS (2)           | 1057.67      | 57.68           | 2.4788       | 565.24     | 23.15 |
> >
> > [1] Kolesnikov V, Matania N, Pinkas B, et al. Practical multi-party private set intersection from symmetric-key techniques[C]//Proceedings of the 2017 ACM SIGSAC Conference on Computer and Communications Security. 2017: 1257-1272.

---

> > > ### Author Response · Authors · 2025-08-01
> > > **Apology Letter**
> > >
> > > Dear Reviewer Uesg,
> > >
> > > We sincerely apologize for submitting our rebuttal during the discussion phase. Due to what appeared to be a delay in receiving the review comments, possibly related to system load on OpenReview, we became aware of the review comments approximately one day after the official NeurIPS release time, which shortened our available rebuttal period. In response to the reviewers’ thoughtful comments, particularly regarding multi-party scaling and other technical aspects, we made preliminary efforts to strengthen our work through code refactoring and additional experiments. We aimed to provide more concrete evidence to address their concerns, though this required more time than anticipated. We truly appreciate the reviewers’ careful feedback and the opportunity to respond. We hope the updated rebuttal is still helpful during the discussion phase, and we remain fully committed to addressing any further questions or suggestions they may have. We strictly adhere to the word limit in the rebuttal rules for rebuttal.
> > >
> > > Thank you for your understanding!
> > >
> > > Best Regards,
> > >
> > > The Authors

---

> ### Comment · Reviewer_Uesg · 2025-08-03
> **Thank for your rebuttal**
>
> Thank you for your response, which addressed most of my concerns, especially regarding the transition from two-party to multi-party settings. This makes me really appreciate the practical nature of this work, so I plan to raise my score to 5.
>
>
> Aside from that, I have a small question: when you mentioned DP, you meant a dynamic programming algorithm, right? Not differential privacy? Since you already ensure privacy with PSI, there’s no need for differential privacy, correct?

---

> > ### Author Response · Authors · 2025-08-03
> > **Thanks for Reviewer's Response**
> >
> > Thank you for your response. We are pleased that this rebuttal addresses most of your concerns. We will address your remaining concerns in the following three areas:
> >
> > 1. We apologize for misleading you by our repeated use of acronyms. In the above rebuttal, "DP" refers to the dynamic programming algorithm, not differential privacy. However, in the context of this paper, the "DP" mentioned refers to Differential Privacy, not Dynamic Programming.
> >
> > 2. Distinction between "DP" in the paper
> > - **Differential Privacy (DP)** is the primary reference for "DP" in the paper. It is explicitly discussed as a security protocol to protect privacy during the transmission of intermediate embeddings. For example:
> >   - In the abstract, it is stated that PubSub-VFL is "compatible with security protocols such as differential privacy."
> >   - Section 4.1 specifies that "we adopt Gaussian Differential Privacy (GDP)... to safeguard the embedding information" to prevent attackers from inferring labels or features from embeddings.
> >   - Appendix C is dedicated to "Differential Privacy Protocol," detailing the Gaussian Differential Privacy (GDP) design and its application in embedding protection.
> >
> > - **Dynamic Programming** is explicitly referred to as "Dynamic Programming" in the paper (e.g., Section 4.3: "Dynamic Programming Algorithm Design") when discussing the optimization of hyperparameters (number of workers, batch size, etc.) and is not abbreviated as "DP."
> >
> >
> > 3. Why Differential Privacy is still needed despite PSI
> > - **PSI (Private Set Intersection)** is used in the paper for secure ID alignment, ensuring that both parties can obtain the "shared" (or aligned) dataset without revealing private information about non-overlapping samples. However, PSI only addresses the privacy of the ID alignment process.
> > - After ID alignment, during model training, the passive party ($P_p$) needs to send intermediate embeddings to the active party ($P_a$). These embeddings may leak sensitive information (e.g., features or indirect label inference) to attackers. Thus, **differential privacy is introduced to add controlled noise to the embeddings**, preventing such inference attacks while balancing privacy and model utility (as explained in Section 4.1 and Appendix C).
> >
> > In summary, "DP" in the paper refers to Differential Privacy, which complements PSI to enhance privacy protection for intermediate data during training, rather than being redundant.
> >
> > If you have any further concerns, please feel free to let us know. We will do our best to address your concerns.

---

> > > ### Comment · Reviewer_Uesg · 2025-08-04
> > > **Response to Authors' Further Reply**
> > >
> > > I see, that makes sense. Thank you for explaining your DP in more detail—it really gave me a sense of the completeness of your work and how carefully you designed the algorithms.
> > >
> > > This brings me to a new question: what is the time complexity of your DP?

---

> > > > ### Author Response · Authors · 2025-08-04
> > > > **Thanks for your response**
> > > >
> > > > We appreciate your prompt response and additional comments.
> > > >
> > > > Given that the dynamic programming algorithm operates over a discrete and bounded search space, its computational complexity is polynomial and tractable. Furthermore, this algorithm operates during the system planning phase (before training), eliminating runtime overhead. It does not interfere with training dynamics, making it suitable for resource-constrained environments. We give the time complexity analysis as follows:
> > > >
> > > > - The dynamic programming algorithm in PubSub-VFL for hyperparameter optimization exhibits a polynomial time complexity, primarily determined by the size of its state space. The state is defined by a triplet $(i, j, r)$, where $i$ indexes the number of active-party workers ($w_a \in [P, Q]$), $j$ indexes passive-party workers ($w_p \in [M, N]$), and $r$ indexes feasible batch sizes $B \in \{B_1, ..., B_R\}$ with $B \leq B_{\text{max}}$. The total number of states is $(Q-P+1) \times (N-M+1) \times R$, and each state involves computing the cost function via $\mathcal{O}(1)$ arithmetic operations (e.g., calculating forward/backward delays and communication costs). Thus, the overall complexity is $\mathcal{O}((Q-P+1) \times (N-M+1) \times R)$, which remains manageable in practice due to constrained ranges for workers (e.g., $[2, 50]$) and batch sizes (e.g., $\{16, 32, ..., 1024\}$), ensuring efficiency for offline hyperparameter tuning. Moreover, it is a one-time offline procedure and does not run during training. Therefore, it does not introduce runtime overhead during the actual learning process.

---

> > > > > ### Comment · Reviewer_Uesg · 2025-08-04
> > > > > **Response to Authors' Further Reply2**
> > > > >
> > > > > Thank you for responding so quickly. I understand the time complexity of DP of your paper, which seems that you guys carefully designed the algorithm.
> > > > >
> > > > > Since I’ve already given a score of 5, I’ll decide whether to raise it to 6 based on your rebuttals to everyone else. I’ll take a quick look and make a decision soon.

---

> > > > > > ### Author Response · Authors · 2025-08-04
> > > > > > **Thanks for your support!**
> > > > > >
> > > > > > Thank you for your tremendous support! If you have any concerns about our rebuttals to other reviewers' comments, please feel free to let us know. Thank you again!

---

> > > > > > > ### Comment · Reviewer_Uesg · 2025-08-05
> > > > > > >
> > > > > > > Thank you so much for your prompt response!
> > > > > > >
> > > > > > > Apologies for my late reply—I was busy working on my own paper’s rebuttal.
> > > > > > >
> > > > > > > I’ve just finished reading your responses to all the reviewers’ comments (HZmk,aqzq,AWXK), and I feel that you’ve addressed the concerns from the other three reviewers as well.
> > > > > > >
> > > > > > > After careful consideration, I’ve decided to raise my score to 6, and I hope this helps improve the chances of your paper being accepted.

---

### Official Review · Reviewer_HZmk · 2025-07-02

**Clarity:** 3
**Significance:** 2
**Originality:** 3
**Rating:** 4
**Confidence:** 3

**Summary:**

This paper introduces PubSub-VFL, a novel VFL approach for two-party split learning designed to tackle inefficiencies and computational resource utilization issues arising from synchronous dependencies and heterogeneous resources/data distribution.
PubSub-VFL combines a Publisher/Subscriber architecture with a parameter server for managing hierarchical asynchronous updates and provides an adaptive strategy for handling resource and data heterogeneity.
The proposed method reduces training latency and enhances computational resource utilization. Extensive experiments across multiple benchmark datasets demonstrate that PubSub-VFL substantially improves computational efficiency (2-7× faster training), achieving state-of-the-art results regarding performance, resource utilization, and privacy guarantees.

**Questions:**

1. The resource and data heterogeneity you mentioned frequently appear as motivating factors. Could the authors clarify how explicitly addressing a two-party scenario aligns distinctly with addressing resource and data heterogeneity?
2. Can the authors provide further reasons for not explicitly comparing with advanced AVFL frameworks like the ones explicitly mentioned in the paper ([36-39])? How would the proposed method's performance be expected to compare in practice?
3. Could the authors provide detailed explanations for the observed accuracy improvements (in Tables 1 and 6), given that the primary improvements were purportedly computational efficiency and reduced delays?
4. Given the various asynchronous and Pub/Sub mechanisms extensively detailed, how easily can PubSub-VFL scale to multi-party scenarios beyond two-party splits?
5. Is the proposed optimization model (equations and dynamic programming solution) computationally lightweight and practical for real-time training scenarios, or would it introduce considerable overhead during initialization phases?

**Ethical Concerns:**

["NO or VERY MINOR ethics concerns only"]

**Limitations:**

yes, in the applendix

**Quality:**

2

**Strengths And Weaknesses:**

Strengths:
- The proposed Pub/Sub architecture combined with parameter servers provides a new solution and a clear design advantage compared to traditional synchronous or conventional asynchronous vertical federated learning frameworks.
- The paper supports its claims extensively through a thorough empirical evaluation across various benchmarks. These results are clear evidence of real-world relevance.
- The authors provide theoretical convergence analyses under the stated assumptions, formally demonstrating that their method converges reliably.

Weaknesses:
- One stated motivation—resource and data heterogeneity—is more relevant in multi-party (rather than two-party) scenarios. Since this paper focuses explicitly on two-party vertical federated learning scenarios, the impact of this "heterogeneity" motivation needs better alignment with demonstration.
- Although the authors explicitly mention leading asynchronous VFL (AVFL) works [36-39], the comparisons in Table 4 (appendix) and the primary tables in the paper lack evaluation against them, leading to insufficient experimental contrast with the actual AVFL state-of-the-art.
- Baseline methods do not sufficiently represent recent AVFL approaches, especially those referred to explicitly, such as [39].
- The main contribution addresses computational efficiency, training latency, and resource utilization. However, Table 1 and Table 6 indicate nontrivial accuracy improvements without a detailed explanation, somewhat conflicting or not clearly aligned with the stated contributions.

---

> ### Author Response · Authors · 2025-08-01
> **Rebuttal**
>
> **W1&Q1:** While resource and data heterogeneity are often associated with multi-party federated learning, this paper demonstrates that such heterogeneity remains highly relevant even in two-party VFL scenarios. We justify this by showing that computational resource disparities (e.g., CPU core ratios like 50:14) and imbalanced feature dimensions (e.g., 50:450) between the two parties can significantly impact training latency and CPU utilization. To address this, the proposed PubSub-VFL explicitly models and optimizes for these imbalances through a privacy-preserving hyperparameter tuning strategy that adapts to each party's system profile. Therefore, the focus on two-party VFL is not a limitation but a targeted demonstration that resource and data heterogeneity can still severely affect efficiency, even in minimal collaborative settings, and merit dedicated architectural and algorithmic interventions.
>
> **W2-3&Q2:** In this paper, we aim to demonstrate the advantages of PubSub-VFL by comparing the performance, resource utilization, and communication overhead of different abstract VFL frameworks. Although abstract AVFL frameworks may involve different implementations (such as [36-39]), their core concept remains the design of asynchronous mechanisms. To further illustrate this point, we follow the experimental setup of the original manuscript and conduct a set of comparative experiments on the Blog dataset, as shown in the table below. The experimental results show that PubSub-VFL still has advantages over the frameworks in [36-39].
>
> | Method | Running Time | CPU Utilization | Waiting Time | Comm. Cost |
> |--------|--------------|-----------------|--------------|------------|
> | [36]   | 952.24       | 24.57           | 12.6514      | 896.34     |
> | [37]   | 1136.36      | 32.35           | 13.3575      | 845.71     |
> | [38]   | 1002.67      | 28.14           | 11.2544      | 924.04     |
> | [39]   | 914.74       | 29.36           | 10.3657      | 887.41     |
> | Ours   | 92.54        | 91.07           | 1.1389       | 439.45     |
>
> **W4&Q3:** While the primary goal of PubSub-VFL is to improve computational efficiency through hierarchical asynchronous design and system-profile-based optimization, the observed accuracy improvements in Tables 1 and 6 are not incidental. These gains can be attributed to several factors:
>
> 1. **Reduced Waiting Time and Staleness**: By eliminating inter-party and intra-party synchronization bottlenecks, the system reduces idle time and stale updates, allowing for more timely and consistent parameter updates. This helps improve convergence stability, especially in heterogeneous environments.
>
> 2. **Better Hyperparameter Configuration**: The optimization of batch size, number of workers, and resource allocation via system profiling contributes to more effective training dynamics, improving generalization and accuracy. As shown in Tables 2 and 3, different configurations significantly affect model performance.
>
> 3. **Smoother Training Process**: The semi-asynchronous mechanism adaptively controls the synchronization interval (∆T), enabling faster convergence in early stages and stable fine-tuning in later stages. This dynamic balance may indirectly enhance model performance.
>
> 4. **Improved Utilization of Data and Compute Resources**: Especially under resource or data heterogeneity, the proposed method enables better use of all available resources, reducing convergence variance that could negatively affect accuracy in baselines.
>
> **Q5:** Given that the dynamic programming (DP) algorithm operates over a discrete and bounded search space, its computational complexity is polynomial and tractable. We give the specific analysis as follows:
>
> - The DP algorithm in PubSub-VFL for hyperparameter optimization exhibits a polynomial time complexity, primarily determined by the size of its state space. The state is defined by a triplet $(i, j, r)$, where \(i\) indexes the number of active-party workers ($w_a \in [P, Q]$), $j$ indexes passive-party workers ($w_p \in [M, N]$), and $r$ indexes feasible batch sizes $B \in \{B_1, ..., B_R\}$ with $B \leq B_{\text{max}}$. The total number of states is $(Q-P+1) \times (N-M+1) \times R$, and each state involves computing the cost function via $O(1)$ arithmetic operations (e.g., calculating forward/backward delays and communication costs). Thus, the overall complexity is $O((Q-P+1) \times (N-M+1) \times R)$, which remains manageable in practice due to constrained ranges for workers (e.g., $[2, 50]$) and batch sizes (e.g., $\{16, 32, ..., 1024\}$), ensuring efficiency for offline hyperparameter tuning. Moreover, it is a one-time offline procedure and does not run during training. Therefore, it does not introduce runtime overhead during the actual learning process.

---

> > ### Author Response · Authors · 2025-08-01
> > **Rebuttal**
> >
> > **Q4:** While PubSub-VFL is currently designed and evaluated in a two-party VFL setting, its core architectural features suggest potential for extension to multi-party scenarios. The decoupled Publisher/Subscriber mechanism inherently supports many-to-many communication patterns, which is advantageous for scaling. Similarly, the hierarchical asynchronous design and buffering strategies can generalize to handle diverse update timings from multiple parties. However, the core challenges are the complexity of ID alignment and the adaptation of optimization algorithms. To address these two challenges, we provide the following two insights:
> >
> > 1. **Complexity of ID Alignment:** To address this challenge, we can leverage existing multi-party PSI techniques (e.g., [1]), which can be applied during the system configuration phase.
> > 2. **Adaptation of Optimization Algorithms:** Because multiple parties are involved, the DP algorithm's search space becomes large, making it difficult to find the optimal solution. To address this challenge, a straightforward approach is to jointly model the passive party with the least resources (known from system profile information) and the active party to determine the optimal hyperparameter configuration. The key insight from doing so is that the key bottleneck dragging down system efficiency is the efficiency gap between the active party and the passive party with the least resources. This idea is consistent with the original manuscript. Although this approach is not optimal, it remains an option for expansion. In this way, our framework can be straightforwardly extended to multi-party scenarios based on the original manuscript.
> >
> > The reason we did not include experiments in multi-party settings in the initial version is due to the focused scope of the research topic and the intended application scenarios. We believe that incorporating both settings in a single paper might dilute the clarity and focus of the core contributions. To implement these improvements, we refactored the PubSub-VFL implementation by modifying several core components. Specifically, we improved the cache mechanism by increasing cache capacity to support more stable training, extended the wait deadline (Tddl) to 15 seconds to enhance the reliability of embedding matching, and refined the dynamic programming optimization algorithm. All other system mechanisms were kept unchanged. Furthermore, we conducted a series of comparative experiments on the Blog dataset to evaluate the performance of PubSub-VFL in multi-party scenarios. The results are summarized in the table below.
> >
> > | Method (# of Parties) | Running Time | CPU Utilization | Waiting Time | Comm. Cost | RMSE  |
> > |-----------------------|--------------|-----------------|--------------|------------|-------|
> > | PubSub-VFL (10)       | 141.14       | 86.32           | 1.9273       | 896.34     | 23.44 |
> > | PubSub-VFL (8)        | 121.55       | 88.36           | 2.0147       | 684.71     | 22.61 |
> > | PubSub-VFL (6)        | 118.36       | 85.69           | 1.5697       | 645.34     | 22.34 |
> > | PubSub-VFL (4)        | 104.72       | 90.14           | 1.2254       | 569.65     | 23.17 |
> > | PubSub-VFL (2)        | 92.54        | 91.07           | 1.1389       | 439.45     | 22.34 |
> > | VFL-PS (10)           | 1324.71      | 52.24           | 1.4410       | 1264.64    | 24.19 |
> > | VFL-PS (8)            | 1374.63      | 47.64           | 1.2147       | 1165.17    | 22.61 |
> > | VFL-PS (6)            | 1245.94      | 50.36           | 1.1647       | 1211.37    | 22.35 |
> > | VFL-PS (4)            | 1174.65      | 51.24           | 1.4211       | 1089.64    | 23.19 |
> > | VFL-PS (2)            | 974.65       | 41.47           | 1.2765       | 874.55     | 23.07 |
> > | AVFL (10)             | 1445.28      | 27.65           | 20.3677      | 1024.34    | 24.54 |
> > | AVFL (8)              | 1274.57      | 28.41           | 21.4154      | 967.57     | 23.71 |
> > | AVFL (6)              | 1198.18      | 28.67           | 17.6517      | 915.16     | 24.01 |
> > | AVFL (4)              | 1181.14      | 25.63           | 16.7456      | 847.65     | 22.84 |
> > | AVFL (2)              | 1068.88      | 21.74           | 15.3657      | 754.77     | 22.97 |
> > | AVFL-PS (10)          | 1274.51      | 67.51           | 2.6971       | 965.59     | 23.08 |
> > | AVFL-PS (8)           | 1165.33      | 68.14           | 2.8146       | 817.55     | 23.67 |
> > | AVFL-PS (6)           | 1017.82      | 58.59           | 2.6511       | 721.38     | 23.61 |
> > | AVFL-PS (4)           | 1197.53      | 61.23           | 2.5636       | 617.45     | 24.07 |
> > | AVFL-PS (2)           | 1057.67      | 57.68           | 2.4788       | 565.24     | 23.15 |
> >
> > [1] Kolesnikov V, Matania N, Pinkas B, et al. Practical multi-party private set intersection from symmetric-key techniques[C]//Proceedings of the 2017 ACM SIGSAC Conference on Computer and Communications Security. 2017: 1257-1272.

---

> > > ### Author Response · Authors · 2025-08-01
> > > **Apology Letter**
> > >
> > > Dear Reviewer HZmk,
> > >
> > > We sincerely apologize for submitting our rebuttal during the discussion phase. Due to what appeared to be a delay in receiving the review comments, possibly related to system load on OpenReview, we became aware of the review comments approximately one day after the official NeurIPS release time, which shortened our available rebuttal period. In response to the reviewers’ thoughtful comments, particularly regarding multi-party scaling and other technical aspects, we made preliminary efforts to strengthen our work through code refactoring and additional experiments. We aimed to provide more concrete evidence to address their concerns, though this required more time than anticipated. We truly appreciate the reviewers’ careful feedback and the opportunity to respond. We hope the updated rebuttal is still helpful during the discussion phase, and we remain fully committed to addressing any further questions or suggestions they may have. We strictly adhere to the word limit in the rebuttal rules for rebuttal.
> > >
> > > Thank you for your understanding!
> > >
> > > Best Regards,
> > >
> > > The Authors

---

> > > ### Comment · Reviewer_HZmk · 2025-08-04
> > > **Response to Authors' Rebuttal**
> > >
> > > Thank you for your comprehensive and thoughtful response to my comments. I appreciate the additional experiments, detailed explanations, and efforts to address my concerns.
> > >
> > > One more question. Your detailed explanation attributing accuracy gains to reduced waiting time, better hyperparameter configuration, smoother training through dynamic synchronization, and improved resource utilization is logical and aligns with PubSub-VFL’s design goals. However, these points would be more convincing if supported by quantitative evidence, such as an ablation study isolating each factor's contribution to the accuracy improvements.

---

> > > > ### Author Response · Authors · 2025-08-04
> > > > **Ablation experiment results**
> > > >
> > > > Thank you for your prompt response and additional comments. We understand that your primary concern is to know which key components (or factors) contribute to improving PubSub-VFL performance. To answer this question, we will address it from the following three perspectives:
> > > >
> > > > - **Identify the Key Factors in Ablation Experiments:** In our previous rebuttal, we highlighted four key factors contributing to the performance gains of PubSub-VFL: reduced waiting time, improved hyperparameter configuration, smoother training via dynamic synchronization, and enhanced resource utilization. These advantages are directly influenced by the four core components of PubSub-VFL, i.e., *the waiting deadline mechanism, the dynamic programming algorithm, the intra-party semi-asynchronous mechanism, and the PubSub architecture*. To systematically evaluate the contribution of each component, we conduct ablation studies that isolate and assess the impact of these elements on the overall system performance.
> > > >
> > > > - **Ablation Studies.** We conduct comprehensive ablation studies following the experimental setup of the original paper, including the same model architectures, number of clients, and heterogeneous resource settings. Experiments are performed on the Energy, Blog, Bank, Credit, and Synthetic datasets to evaluate the individual contributions of PubSub-VFL’s key components. Specifically, to assess the impact of the waiting deadline mechanism, we set the deadline to $T_{\text{ddl}} = 0$s, effectively disabling the mechanism. To evaluate the dynamic programming algorithm, we adopt a fixed worker allocation (i.e., equal numbers of workers on both party sides), removing the adaptive scheduling it enables. For the intra-party semi-asynchronous mechanism, we remove this component while retaining the PS architecture to isolate its effect. Finally, to study the role of the PubSub architecture, we replace it with the AVFL-PS architecture while keeping all other components unchanged. These ablation experiments allow us to disentangle the influence of each design choice on overall system performance. The experimental results are shown in the following table.
> > > >
> > > > | Method                     | Energy | Blog  | Bank  | Credit | Synthetic |
> > > > |----------------------------|--------|-------|-------|--------|-----------|
> > > > | All (PubSub-VFL)           | 83.94  | 22.14 | 96.97 | 86.07  | 94.17     |
> > > > | w/o $T_{ddl}$              | 84.35  | 23.17 | 95.26 | 85.74  | 92.86     |
> > > > | w/o Dynamic Programming    | 84.07  | 22.16 | 96.33 | 85.79  | 93.82     |
> > > > | w/o $\Delta T$             | 85.68  | 24.11 | 95.01 | 84.45  | 92.07     |
> > > > | w/o PubSub                 | 83.98  | 22.66 | 95.17 | 85.93  | 93.52     |
> > > > | w/o $T_{ddl}$ and $\Delta$ | 85.81  | 24.24 | 94.32 | 82.69  | 91.73     |
> > > > | VFL                        | 84.24  | 23.18 | 94.97 | 83.42  | 92.74     |
> > > > | VFL-PS                     | 86.14  | 23.07 | 94.74 | 85.44  | 92.67     |
> > > > | AVFL                       | 83.91  | 22.97 | 95.02 | 84.23  | 91.54     |
> > > > | AVFL-PS                    | 84.29  | 23.15 | 95.06 | 82.27  | 92.21     |
> > > >
> > > >
> > > > - **Key Observations:** We summarize our key findings from the ablation studies as follows:
> > > >
> > > > 1. The Waiting Deadline Mechanism and the Intra-party Semi-asynchronous Mechanism contribute most significantly to the performance gains of PubSub-VFL. This is expected, as these components play a critical role in smoothing the training process and mitigating gradient staleness by balancing synchronization and asynchrony. For instance, removing the waiting deadline mechanism ($T_{ddl}$) leads to noticeable performance drops across all datasets: the AUC on the Bank dataset decreases by 1.71% (from 96.97% to 95.26%), and the RMSE on the Blog dataset increases by 1.03 (from 22.14 to 23.17). Similarly, omitting the dynamic synchronization interval ($\Delta T$) results in even more severe degradation: the Synthetic dataset’s AUC falls by 2.10% (from 94.17% to 92.07%), and the Credit dataset’s AUC drops by 1.62% (from 86.07% to 84.45%). Their combined effect ensures timely updates and reduces idle time without compromising convergence stability.
> > > >
> > > > 2. The PubSub architecture and the dynamic programming algorithm, while contributing less directly to raw performance improvement, enhance training stability and convergence robustness. Removing the PubSub architecture causes moderate performance declines, such as a 1.80% drop in Bank dataset AUC (from 96.97% to 95.17%) and a 0.65% decrease in Synthetic dataset AUC (from 94.17% to 93.52%). Without dynamic programming, the Bank dataset’s AUC decreases by 0.64% (from 96.97% to 96.33%), and the Credit dataset’s AUC falls by 0.28% (from 86.07% to 85.79%). These results indicate that they help alleviate issues arising from heterogeneous client resources and reduce interference in coupled training dynamics. By better managing resource imbalances and coordination overhead, these components foster a more stable and resilient asynchronous training process.

---

> > > > > ### Author Response · Authors · 2025-08-04
> > > > > **Thanks**
> > > > >
> > > > > Dear Reviewer HZmk,
> > > > >
> > > > > Thank you for your constructive review comments.
> > > > >
> > > > > If you have any additional concerns, please feel free to let us know. We will do our best to address your concerns. If you are satisfied with our rebuttal and explanation, we would be grateful if you could adjust your rating appropriately.
> > > > >
> > > > > Best regards,
> > > > >
> > > > > The Authors

---

> > > > > > ### Comment · Reviewer_HZmk · 2025-08-06
> > > > > >
> > > > > > Thank you to the authors for their detailed response, which addressed most of my concerns. Given the significance and novelty of this work, I will maintain my positive score.

---

> > > > > > > ### Author Response · Authors · 2025-08-07
> > > > > > > **Thanks for your reply!**
> > > > > > >
> > > > > > > Dear Reviewer HZmk,
> > > > > > >
> > > > > > > Thank you for your prompt response and constructive comments! We promise to incorporate the above changes into future revisions. In addition, we remind you that you need to click the "edit" button in the review comments box and enter a "final rating," in accordance with the rules and policies in "ACK." Thank you again for your tremendous support!
> > > > > > >
> > > > > > > Best regards,
> > > > > > >
> > > > > > > The Authors

---

### Comment · Area_Chair_gHxk · 2025-08-09
**Author-review discussion and mandatory acknowledgment**

Dear reviewers,

Thank you for your efforts and participating in discussion with the authors. We have a few hours remaining for author-reviewer discussion. If not already arrived at a conclusion in your discussions, can we please finalize now?

Please remember to complete the mandatory acknowledgment after having responded to the authors on concluding the discussion.

Thanks,
AC

---

### Note · Authors · 2025-08-12

Dear SAC, AC, and All Reviewers,

To facilitate your final decision, we summarize the details of the rebuttal below:
# Summary of Reviewers' Key Praises
The work was highly regarded by reviewers: HZmk highlighted the novel integration of PubSub with PS and thorough empirical and theoretical validation; Uesg commended its originality, solid theoretical derivations, and in-depth analysis of resource heterogeneity; aqzq appreciated the asynchronous design enabling flexible collaboration and experimentally supported claims; and AWXK acknowledged the careful system design and effective fusion of PubSub and asynchrony.
# Resolution of Most Concerns
We have addressed most of the concerns through supplementary experiments, detailed explanations, and theoretical analysis:
- For Reviewer HZmk: We added comparative experiments with AVFL frameworks on the Blog dataset, showing PubSub-VFL’s superiority in runtime and resource utilization; explained accuracy gains via reduced staleness, optimized hyperparameters, and smoother training; demonstrated multi-party scalability with refactored components and experimental results; and proved the optimization model’s polynomial complexity (tractable for offline tuning).
- For Reviewer Uesg: We evaluated scalability on the Criteo 1TB dataset, validating efficiency; detailed the intra-party semi-asynchronous mechanism’s motivation with adaptive synchronization intervals; and presented multi-party experiments showing consistent performance gains.
- For Reviewer aqzq: We clarified worker selection is an offline optimization (not dynamic during training) problem; explained Eq. (5)’s design logic and overfitting mitigation; derived Eq. (6) in detail, correcting notation ambiguities; and confirmed system profiles (infrastructure metadata) pose minimal privacy risks.
- For Reviewer AWXK: We addressed novelty by emphasizing system-level integration tailored to two-party VFL, validated two-party relevance with industrial use cases, and provided multi-party extension strategies. However, since residual concerns remain (e.g., deeper theoretical analysis for non-convex settings), we respectfully request specific details to further refine the revised version.

Our contributions (PubSub-VFL) have been affirmed by reviewers. The resolved concerns, supported by supplementary experiments and analyses, strengthen the work’s rigor. We kindly ask SAC and AC to consider these efforts when making the final decision.

Thanks,

The Authors

---

### Decision · Program_Chairs · 2025-09-17

**Decision:**

Accept (poster)

**Comment:**

The submission proposes a novel method for vertical federated learning to handle limitations from synchronous dependencies and heterogenous data distributions. HZmk requires more quantitative results to support the advantages of dynamic synchronization, which are provided in the author response and deemed satisfactory. Uesg strongly supports the careful design of the proposed method, while questioning scalability and relevance of the two-party setting, which are adequately addressed in the author rebuttal by including additional experiments and explaining real-world setting where the two-party setting is currently deployed. aqzq raises concerns on the definition of computational delay for active and passive parties, which requires additional analysis and improvements in clarifying the assumptions in the author response, which are deemed acceptable by the reviewer. AWXK requires consideration of multi-party scenarios and utility degradation at low privacy budgets, which are addressed by the rebuttal but did not entirely convince the reviewer, who does slightly raise the score. The AC notes that similar questions are raised by Uesg too and the answers are satisfactory in justifying the importance of the two-party setting in itself, while Figure 5 is an adequate analysis on impact of privacy budget. Overall, the paper proposes a thorough method to solve a relevant problem with a novel approach, thus, is recommended for acceptance. The authors are encouraged to include the clarified definitions and additional experiments from the discussion in the final version.